# Bacterial Endophytes as a Promising Approach to Enhance the Growth and Accumulation of Bioactive Metabolites of Three Species of *Chenopodium* Sprouts

**DOI:** 10.3390/plants10122745

**Published:** 2021-12-13

**Authors:** Mohammed S. Almuhayawi, Mohamed Abdel-Mawgoud, Soad K. Al Jaouni, Saad M. Almuhayawi, Mohammed H. Alruhaili, Samy Selim, Hamada AbdElgawad

**Affiliations:** 1Department of Medical Microbiology and Parasitology, Faculty of Medicine, King Abdulaziz University, Jeddah 21589, Saudi Arabia; malruhaili@kau.edu.sa; 2Department of Medicinal and Aromatic Plants, Desert Research Centre, Cairo 11753, Egypt; 3Hematology/Pediatric Oncology, Yousef Abdulatif Jameel Scientific Chair of Prophetic Medicine Application, Faculty of Medicine, King Abdulaziz University, Jeddah 21589, Saudi Arabia; saljaouni@kau.edu.sa; 4Department of Otolaryngology-Head and Neck Surgery, Faculty of Medicine, King Abdulaziz University, Jeddah 21589, Saudi Arabia; salmehawi@kau.edu.sa; 5Department of Clinical Laboratory Sciences, College of Applied Medical Sciences, Jouf University, Sakaka 72388, Saudi Arabia; sabdulsalam@ju.edu.sa; 6Department of Botany and Microbiology, Faculty of Science, Beni-Suef University, Beni-Suef 62521, Egypt; hamada.abdElgawad@uantwerpen.be

**Keywords:** bacterial endophytes, *Chenopodium* sp., sprouts, photosynthesis, amino acid metabolism, phenolics metabolism, antioxidant, anti-inflammatory

## Abstract

Sprouts are regarded as an untapped source of bioactive components that display various biological properties. Endophytic bacterium inoculation can enhance plant chemical composition and improve its nutritional quality. Herein, six endophytes (Endo 1 to Endo 6) were isolated from *Chenopodium* plants and morphologically and biochemically identified. Then, the most active isolate Endo 2 (strain JSA11) was employed to enhance the growth and nutritive value of the sprouts of three *Chenopodium* species, i.e., *C. ambrosoides*, *C. ficifolium,* and *C. botrys*. Endo 2 (strain JSA11) induced photosynthesis and the mineral uptake, which can explain the high biomass accumulation. Endo 2 (strain JSA11) improved the nutritive values of the treated sprouts through bioactive metabolite (antioxidants, vitamins, unsaturated fatty acid, and essential amino acids) accumulation. These increases were correlated with increased amino acid levels and phenolic metabolism. Consequently, the antioxidant activity of the Endo 2 (strain JSA11)-treated *Chenopodium* sprouts was enhanced. Moreover, Endo 2 (strain JSA11) increased the antibacterial activity against several pathogenic bacteria and the anti-inflammatory activities as evidenced by the reduced activity of cyclooxygenase and lipoxygenase. Overall, the Endo 2 (strain JSA11) treatment is a successful technique to enhance the bioactive contents and biological properties of *Chenopodium* sprouts.

## 1. Introduction

Sprouts have been recognized as excellent sources of bioactive phytochemicals, such as proteins, vitamins, minerals, and phenolic compounds, which exhibit a variety of nutritive and health promoting characteristics, e.g., anticancer and antioxidative activities [1,2]. The reduced levels of antinutritional factors also give the sprouts additional advantages over seeds and mature plants, thus, sprouts could be qualified as natural healthy foods [2]. Therefore, the application of various techniques for improving the nutraceutical and functional food values of sprouts and their contents of bioactive metabolites has recently been a research hotspot [3,4]

Bacterial endophytes are a large group of microorganisms colonizing the internal plant tissues, mainly in the roots. Such a symbiotic relationship between plants and endophytic bacteria is supposed to lead to a set of biochemical and physiological changes, which eventually offer many benefits to the host plant, either directly or indirectly [5]. The direct effect of endophytes includes the production of growth regulators, phosphate solubilization, and nitrogen fixation, while the indirect effect occurs in response to pathogen infection to enhance the plant’s resistance against diseases [5,6,7]. The production of siderophore by endophytic bacteria could also protect the plant against phytopathogens through reducing the availability of iron for some pathogens [8]. Bacterial endophytes have also been known to be rich sources of bioactive secondary metabolites [9]. Therefore, bacterial endophytes could play a significant role in plant growth, by induction of such growth promoting effects on plants, which recommend them as excellent candidates for improving plant yield, as well as the accumulation of bioactive compounds [10,11]. For instance, a positive effect of bacterial strains on the bioactive content of sesquiterpenoid of *Atractylodes lancea* was previously reported [11]. In addition, bacterial endophytes have been utilized to improve the quality and nutritive value of *Lyophyllum decastes* by enhancing its bioactive components [12]. However, few attempts have been made to explore the bacterial endophyte-induced changes in the plant metabolome. Therefore, detailed metabolic studies are important in order to understand (i) the changes associated with the plant–endophyte interaction and (ii) how the plant’s bioactive metabolites respond to such an interaction to improve a plant’s nutritional and pharmaceutical values.

*Chenopodium* species belong to the *Amaranthaceae* family. Their seeds and sprouts have been known for their high nutritive values. *Chenopodium* sprouts are considered as sources for proteins, vitamins, antioixdants, and carbohydrates, e.g., *Chenopodium quinoa* [13,14] and *Chenopodium formosanum* Koidz. [15]. Sprouting has been reported to induce modifications in storage proteins in quinoa, with a concomitant increase in some metals, such as Cu and Zn [13]. For instance, *Chenopodium ambrosioides* has been traditionally used for the treatment of bronchitis, tuberculosis, vomiting, and skin ulcerations. It has also anthelmintic, anti-inflammatory, antipyretic, and analgesic properties [16]. In addition, *C. ambrosioides* has been considered as a rich source of carotenoids, proteins, and fats [17]. Similarly, *C. botrys* has been used for treating coughs and abdominal pain. It has also some medicinal properties, such as antidiuretic, antispasmodic, anticonvulsant, anti-inflammatory, antidiabetic, antibacterial, antifungal, and antiviral [17]. *C. botrys* also contains a variety of bioactive compounds, mainly flavonoids, anthraquinones, saponins, and tannins [18].Therefore, enhancing the phytochemical content of such medicinal plants and their sprouts by using ecofriendly approaches, such as bacterial endophytes, could increase their health promoting and nutritive values. For instance, calcium as an essential nutrient for plants was applied to improve growth, antioxidant levels, and nutrient availability in *Chenopodium formosanum* sprouts [15]. Similarly, high CO_2_ was applied to increase the growth of *Chenopodium album* sprouts [19]. This ecofriendly approach could also support their traditional uses as well as their application in pharmaceutical industries and also to ensure safety to the environment. To our knowledge, the influence of endophytic bacteria on the growth and bioactive metabolites of the selected sprouts (*Chenopodium ambrosoides, Chenopodium ficifolium,* and *Chenopodium botrys*) has not been previously investigated. In addition, endophytic bacteria have not been intensively studied when compared with endophytic fungi [9,20]. Thus, the present study was conducted with the aim to isolate and identify a bioactive endophyte to improve *Chenopodium* growth and tissue quality. For this purpose, we investigated the effects induced by the endophytic bacteria *Streptomyces* on growth, physiology, levels of minerals, and some primary and secondary metabolites in the tested *Chenopodium* sprouts. Additionally, the concomitant biological activities, i.e., antioxidant, antimicrobial, and anti-inflammatory activities were measured. We hypothesize that the use of a bioenhancer, such as endophytic bacteria, could elevate the health promoting and nutritive values of the tested sprouts.

## 2. Results and Discussion

### 2.1. Characterization of the Isolated Endophytic Bacteria

In the current investigation, a total of six endophytic bacterial strains were isolated and characterized (Endo 1 to Endo 6) on the basis of morphological and biochemical traits. Most of the tested endophytic bacteria showed variability in substrate color and aerial mycelia, besides their ability to produce diffusible pigments (Table 1). The aerial hyphae of the examined endophytes were also observed with long spiral spore chains, long rectiflexible spore chains, or verticillate spores, such as those previously reported for other bacterial strains [21]. The morphological examinations of the six isolates indicated that these isolates belong to the genus *Streptomyces*, where its morphologically related genera have extensively branched mycelia [22].

Regarding the biochemical and physiological attributes, most isolated endophytes were capable of utilizing different carbon sources (e.g., galactose and sucrose) as well as nitrogen sources (e.g., cysteine and tyrosine) in addition to their ability to produce antioxidant metabolites (phenolics and tocopherols) and several bioactive enzymes, such as the antioxidant enzymes (catalase and peroxidase), N metabolic enzymes (e.g., L-asparaginase, glutaminase, urease and nitrate reduction), and C metabolic enzymes (e.g., starch and casein hydrolysis and lipolysis). Moreover, endophytic isolates showed a high antioxidant capacity (FRAP, DPPH, and ABTS%) and solubilization of tricalcium phosphate (Table 1). The production of such enzymes could enable the tested bacteria to survive during unfavorable conditions [23]. In this regard, it was found that the utilization of carbon and nitrogen sources by some bacterial strains might affect their production of bioactive secondary metabolites [24].

### 2.2. Selection of the Most Active Endophytic Bacteria

Bacterial endophytes have been considered as rich sources of bioactive secondary metabolites, such as flavonoids, phenolics, alkaloids, and terpenoids [9]. The assumption has also been that the presence of such bioactive compounds in bacterial endophytes might be responsible for their enhanced biological activities [20]. This can significantly contribute to improving plant growth and biological activity, as indicated by previous reports that described the significant impact of bacterial endophytes on enhancing the nutritive value of tartary buckwheat sprouts by increasing their total and individual flavonoid contents, e.g., rutin and quercetin [25]. Interestingly, a positive correlation was observed between the phenolic content and antioxidant activities of all the tested endophytic bacteria. The most active isolate endophyte 2 (Endo 2) was selected for the availability of its carbon and N sources, and their utilization and metabolism (e.g., starch hydrolysis, nitrate reduction, urease, and L-asparaginase) and also for the highest production of phenolic and tocopherol content, antioxidant activities, and tricalcium phosphate solubilization. In this regard, bacterial endophytes that are rich in phenolics and flavonoids showed highly different biological activities that directly or indirectly support plant growth. Moreover, phenolics production supports the endophytes to establish exclusive symbiotic relationships with plants [26].

This strain was further analyzed for its phenolic profile, whereas 13 phenolic acids and flavonoids were quantified, being dominated by isoquercetrin and apigenin. In this regard, bacterial endophytes are known to contain high levels of phenolic compounds, which might account for their biological properties, particularly the antimicrobial activity [27]. In addition, the production of phytohormones, such as gibberellins (GA), abscisic acid (ABA), and indole acetic acid (IAA), as well as siderophore was also confirmed in the selected endophytic bacterium (Endo 2) as an indication for the growth promoting potential of such an endophyte. In this regard, bacterial endophytes have been known to produce GA and IAA, which may exert growth promoting effects on plants [20]. IAA might also induce a protective effect against environmental stress conditions by enhancing various cellular defense mechanisms [28]. More interesting, the production of siderophores by bacterial endophytes could enable them to overcome some adverse environmental conditions and also reduce the availability of iron for some plant pathogens [8]. Some endophytic bacteria, such as *Stenotrophomonas maltophilia,* have been shown to synthesis catechol-type siderophores [29]. Thus, (Endo 2) was selected for improving the nutritive values of the tested sprouts, i.e., *C. ambrosoides, C. ficifolium,* and *C. botrys*.

### 2.3. Molecular Characterization of the Most Active Isolate

According to the 16S rRNA gene sequence analysis, the Endo 2 strain is affiliated within the genus *Streptomyces* (strain JSA11). With a similarity of >97%, it is closely related to unidentified species within the same genus as shown in Figure 1. The 16S rRNA gene data of the actinobacterial strain reported in this study have been deposited in the NCBI and GenBank nucleotide sequence databases under the accession number (MZ489115).

### 2.4. Bacterial Endophytes Promoted Photosynthesis and Biomass Production of Chenopodium Sprouts

Endophytic bacteria have been assumed to promote plant photosynthetic activity by enhancing the chlorophyll content [30]. Supporting such a hypothesis, the present investigation has clearly revealed that the inoculation of the tested sprouts with *Streptomyces* (strain JSA11) has resulted in significant increases in photosynthetic activity and respiration rate, when compared with the control sprouts (Table 2). In this regard, endophytes have been known to have positive impacts on photosynthesis, which consequently could enhance photosynthetic light and carbon reactions [31]. Similar to our results, sugar beet plants inoculated with endophytes have been shown to have a higher dark respiration rate (i.e., higher light saturation point, light compensation point, and photochemical efficiency) than noninoculated plants [30]. The increased photosynthetic capacity might be due to the ability of enodophytic bacteria to produce some compounds which enhance the electron transport system, which in turn could provide the NADPH and ATP needed for carbon assimilation [30].

The results obtained and shown in (Table 2) have also demonstrated the positive impact of *Streptomyces* (strain JSA11) on chlorophyll and pigment contents of the tested sprouts, whereby significant increases in chlorophyll a and b were observed for *C. ficifolium* and *C. botrys* but not for *C. ambrosoides* under endophytic bacterial stimulation. In addition, all the examined species showed a significant enhancement in β-carotene; however, chlorophyll a and b were enhanced only in *C. botrys* when grown under endophytic bacterial inoculation. The increased chlorophyll contents in response to bacterial inoculation could be due to enhancing the chloroplast metabolism [30]. Such a result is in accordance with those previously obtained by [20], who demonstrated that the inoculation of tomato plants with bacterial endophytes led to increments in chlorophyll contents. Moreover, the biosynthesis of IAA by endophytic bacteria has been demonstrated to enhance the production of different pigments and metabolites [32]. Previous findings have dealt with the positive role of inoculating sunflower plants with growth promoting bacteria, which led to enhanced levels of chlorophyll a and b and carotenoids [33]. Increased levels of chlorophyll and photosynthetic pigments were also reported in mustard plants in response to growth promoting bacteria, which could effectively enhance the enzymes responsible for chlorophyll biosynthesis [34]. Thus, endophytic bacteria could lead to a better growth performance of plants through their positive impact on photosynthesis and chlorophyll content.

As a consequence of improved photosynthesis, the biomass production (expressed as FW) of all the *Streptomyces* (strain JSA11)-inoculated sprouts was significantly enhanced when compared with the untreated control plants, whereas *C. ficifolium* appeared to have the highest biomass accumulation under both the control and bacterial inoculation conditions (Figure 2). In addition, the water content of all the tested sprouts was not affected by the bacterial treatments (Figure 2). This may indicate that endophyte treatment increased more dry weight accumulation (e.g., increased primary metabolites such as sugars, proteins, and fatty acids). This was also consistent with the observed increase in photosynthesis.

In accordance, the higher yield of runner bean plants inoculated with rhizobacteria has been found to be associated with higher photosynthetic activities [35]. Consequently, the higher the photosynthetic activity, the more sugar synthesis by plants, which could lead to a higher growth rate. For instance, sugar beet plants inoculated with endophytes have been found to accumulate higher total carbohydrates than the control plants [30]. However, other reports have demonstrated that additional carbohydrates, synthesized by host plants, might be diverted to feed endophytes at the expense of increasing biomass accumulation [31,36]. This could explain the mutualistic association between endophytes and the host plants, whereas the host plants could benefit from an increased growth rate. In return, endophytes could gain more nutrients from the host plant [30].

On the other hand, the growth promoting effects of endophytic bacteria on plants might be ascribed to their ability to produce growth regulators, such as GA and IAA, which consequently stimulate cell elongation and differentiation in plants [20,32]. Moreover, the production of IAA by endophytic bacteria has been shown to improve photosynthesis, which would be positively reflected on the better growth performance of the target plants [32]. For example, bacterial endophytes have been previously reported to induce positive effects on growth and biomass production i.e., higher fresh weight (FW) and dry weight (DW) in buckwheat sprouts [37], as well as many plant species, such as tomato plants [20] and sugarcane plants [30]. Furthermore, other mechanisms triggered by endophytic bacteria might be involved in plant growth promotion, possibly through enhanced nutrient uptake and antagonistic effects against phytopathogens, nitrogen fixation, or the production of phytohormones, siderophores, and secondary metabolites [5,11]. The endophytes-produced organic acids were also assumed to play a role in plant growth promoting traits and the protection against pathogens [5].

### 2.5. Improved Minerals and Vitamins Contents by Endophytic Bacterial Treatment Contribute to Enhancing the Nutritive Value of Chenopodium Sprouts

Endophytic bacteria have been supposed to enhance the nutrient absorption by plants, particularly N, P, K, Ca, and Mg, leading to better nutrient use efficiency [38]. In the present study, the mineral profile of the investigated sprouts was analyzed, whereas nine mineral elements were determined, i.e., K, Na, Ca, Cu, Fe, P, Zn, Mn, and Mg (Table 3). When inoculated with *Streptomyces* (strain JSA11), the tested plants showed marked increases in almost all the detected elements. The percentage of increase in K content was recoded in both the control and inoculated plants. It increased by about 5%, 110%, and 100% in *C. ambrosoides*, *C. ficifolium,* and *C. botrys,* respectively, when compared with the control. The enhanced nutrient uptake by plants under endophytic bacterial treatment could be explained by the ability of bacterial endophytes to decrease the endogenous ethylene, hence allowing more nutrient absorption by increased root growth, in addition to the effective role of IAA, produced by endophytic bacteria, in stimulating the uptake of nutrients by plants [39]. Several reports could support our results and explain the endophytic bacterial-induced increments in minerals. For instance, the contribution of endophytic bacteria to phosphate solubilization processes has been well documented. On the one hand, the release of organic acids converts insoluble phosphates into a soluble form, thus improving P utilization efficiency and making it available for plants and, on the other hand, it also increases soil fertility [40]. The increases in K contents in plants could be explained by the ability of endophytic bacteria to solubilize K salts into more soluble forms, which consequently could be taken up by the plant roots, thus enhancing plant growth [10]. Moreover, the Cu content has been previously reported to increase in *Brassica napus* as a result of bacterial inoculation [41]. Regarding the mineral content of the *Chenopodium* species, it has been previously found that the most abundant macroelement in *C. botrys* was K, followed by Ca, P, and Mg, while the predominant microelement in *C. botrys* was Fe, followed by Na, Mn, Zn, and Cu [42].

The results of the current investigation also indicate that the vitamin C and E contents were significantly enhanced in the three species in response to inoculation with *Streptomyces* (strain JSA11) when compared with the control plants (Table 3). In addition, riboflavin and thiamin only increased in *C. botrys*. Meanwhile, vitamin E had the highest concentration among the detected vitamins. Our results could be supported by previous reports of [43], who found that vitamin E was among the main components of *C. ambrosioides*.

The proximate composition of the examined sprouts showed that the total protein, crude fiber, ash, and carbohydrate contents were remarkably enhanced in both *C. ficifolium* and *C. botrys* inoculated with *Streptomyces* (strain JSA11), while such parameters were not significantly changed in *C. ambrosoides*, except for the total proteins. (Table 3). At the same time, bacterial endophytes caused a significant increment in fat content in both *C. ficifolium* and *C. botrys*. In line with our results, the total sugar and protein contents of mushrooms were found to be enhanced in response to bacterial endophytes, resulting in a higher plant nutritive value [12]. Bacterial treatment has also caused elevations in sugar levels in mustard plants [34]. Further, the growth promoting rhizobacteria have been reported to be potent inducers of the plant nutritive value through increasing total protein and carbohydrate content in *Phaseolus coccineus* [35]. In this regard, the endophytic bacterial-induced increase in plant protein content might be attributed to the higher N fixation by the endophytic bacteria [35].

In the same context, the proximate composition of both *C. ambrosioides* and *C. botrys* (i.e., moisture content, ash content, crude protein, crude fiber, crude fat, and carbohydrates) has been previously analyzed and proven to fall within the standard values for some vegetable drugs, which could support the use of this plant in pharmaceutical preparations [44]. *C. ambrosioides* has been reported to be a rich source of carotenoids, as well as proteins and fats [17]. Overall, the enhanced levels of minerals, vitamins, sugars, and proteins in *Chenopodium* sprouts in response to bacterial endophytes treatments could support the plant nutritive value

### 2.6. Endophytic Bacterial Treatment Improved the Functional Food Value of Chenopodium Sprouts through Enhancing Their Bioactive Primary Metabolites Levels

The enhanced photosynthetic capacity as a result of endophytic bacterial inoculation is supposed to increase the plant’s sugar content, which in turn, could be further utilized to meet the energy required for the production of different types of bioactive metabolites [11].

In the current study, the three tested species showed a qualitatively similar, but quantitatively different amino acid profile (Table 4). Apparently, most of the detected amino acids in all the investigated plants were significantly promoted when inoculated with *Streptomyces* (strain JSA11) when compared with control plants. However, some amino acids remained unaffected when treated with endophytic bacteria, e.g., arginine, alanine, histidine, and valine in the case of *C. ficifolium* and *C. botrys* and alanine and isoleucine in the case of *C. ambrosoides*. Notably, the highest concentrations were reached for leucine, lysine, arginine, glutamic acid, and glutamine in all the tested sprouts. In this regard, the uptake of amino acids by plants was greatly affected by the N fixation by microbial communities [45]. Similar to our findings, bacterial treatment has been previously reported to induce higher concentrations of asparagine, therionine, serine, glutamine, alanine, valine, methionine, leucine, tyrosine, phenyl alanine, lysine, histidine, arginine, and proline in mustard plants [34]. Various amino acids (e.g., histidine, leucine, methionine, and tyrosine) were also significantly increased in mushrooms treated with bacterial endophytes [12]. Moreover, the endophytic bacterial association with sugarcane plants led to an accumulation of some amino acids, such as alanine and glycine, while others (e.g., proline and aspartate) were reduced [38]. Such variability might be dependent on the plant species or growth conditions. The majority of the detected amino acids, such as valine, alanine, leucine, proline, isoleucine, serine, threonine, phenyl alanine, asparagine, and tyrosine were previously reported in *C. ambrosioides* [43].

In order to obtain more insight into the amino acid-related changes under endophytic bacterial treatments, the glutamine synthases (GS), dihydrodipicolinate synthase (DHDPS), and cystathionine γ-synthase (CGS) were evaluated as being the key enzymes involved in N metabolism and the synthesis of glutamine, lysine, and methionine [4]. The results obtained show remarkable increases in such enzymes in the *Streptomyces* (strain JSA11)-treated sprouts when compared with controls (Table 4). Our results could be supported by previous studies that described the increased enzymatic activity of glutathione synthase (GS) in sugarcane plants associated with bacterial endophytes [38]. However, it has been shown that endophytic bacteria decreased the tyrosine aminotransferase activity in the hairy roots of *Salvia miltiorrhiza* [46]. Such a difference might depend on the interacting species or growth conditions.

From the present results, it is also clear that the tested species shared the same fatty acid profile qualitatively but not quantitatively, but pentadecanoic (C16:0) and octadecanoic (C18:2) showed the highest concentrations among the saturated and unsaturated fatty acids, respectively, in all the examined plants (Table 4). Some of the detected fatty acids (e.g., tetradecanoic, docosanoic, and octadecanoic) were significantly enhanced, while others (e.g., eicosanoic, pentacosanoic, and octadecanoic) were not affected in all the *Streptomyces* (strain JSA11)-treated sprouts when compared with the untreated control plants. Supporting our results, it has been previously shown that *C. ambrosioides* had higher levels of unsaturated fatty acids (dominated by linolenic and linoleic) than saturated fatty acids [47]. In addition, several fatty acids have been characterized from *C. ambrosioides*, e.g., palmitic, octadecadienoic, stearic, palmitic, octadecadienoic, and stearic acids [43].

Regarding their content of organic acids, the examined *Chenopodium* sprouts contained five organic acids, i.e., oxalic, malic, succinic, citric, and lactic acids (Table 4). Similarly, *C. ambrosoides* has been previously demonstrated to contain oxalic, ascorbic, quinic, fumaric, malic, lactic, succinic, and citric acids with oxalic acid being the most dominant one [43,47]. The application of *Streptomyces* (strain JSA11) endophytic bacterial treatment did not induce significant elevations in most of the detected organic acids, except for malic acid (in *C. botrys*), oxalic acid (in *C. ficifolium* and *C. botrys*), and lactic and succinic acids (in *C. botrys*). Consistent with our results, the organic acid content of mushrooms (e.g., oxalic, tartaric, formic, malic, and fumaric acids) could be significantly altered by bacterial endophytes to improve the plant’s nutritive value [12].

### 2.7. Endophytic Bacteria Stimulated the Antioxidant Activities of Chenopodium Sprouts through Enhancing Their Phenolic Content

The functional food value of plants has been assumed to be associated with their levels of secondary metabolites [48] that could be enhanced under endophytic bacterial stimulations through their ability to produce IAA, which in turn is likely to be utilized by plants to synthesize sugars. Thus, the increased amount of sugar might be further employed for the production of bioactive secondary metabolites [11]. Additionally, the endophytic bacterial-secreted metabolites were hypothesized to play a role in the production of some plant secondary metabolites [11].

The results obtained herein show that *C. ambrosoides* had higher amounts of total phenolic and flavonoids contents than *C. ficifolium* and *C. botrys* under control conditions (Figure 3). Meanwhile, *Streptomyces* (strain JSA11) bacterial endophytes caused increases in the total flavonoids in all the tested sprouts, as well as increases in total phenolics in only *C. ficifolium* and *C. botrys*. Likewise, high phenolic and flavonoid contents were previously observed for tartary buckwheat sprouts inoculated with growth promoting bacteria [25]. Moreover, *C. ambrosioides* and *C. botrys* were shown to possess high total flavonoids and phenolic contents [49,50].

In order to follow the metabolic pathway responsible for such increases in polyphenols, the enzymatic activity of phenylalanine ammonia lyase (PAL), a key enzyme in polyphenol production, was evaluated [4]. PAL catalyzes the nonoxidative elimination of ammonia from its substrate (phenylalanine) to give trans-cinnamic acid. Bacterial endophytes were proposed to play a significant role in the synthesis of plant secondary metabolites via their regulatory effect on enzyme activities [46]. Our results indicate that the *Streptomyces* (strain JSA11) endophytic bacterial treatment led to significant increases in PAL enzyme activity in both *C. ambrosoides* and *C. botrys*, while significant increments were reported for t substrate phenylalanine in *C. ficifolium* and *C. botrys*, in comparison to the untreated control plants (Figure 3). Similarly, previous studies have investigated the increments in such enzyme in response to eCO_2_ [4]. However, previous reports have shown endophytic bacteria to decrease phenylalanine ammonia lyase activity in the hairy roots of *Salvia miltiorrhiza* [46]. Such variability might be dependent on the plant species or growth conditions. Thus, the regulation of plant secondary metabolites by endophytic bacteria might be governed by their impact on enzymatic activity.

The increments in the phenolic content in the studied species in response to the *Streptomyces* (strain JSA11) inoculation were also observed to be concomitant with dramatic increases in antioxidant activities (evaluated by FRAP, ABTS, and DPPH assays) (Table 5). Meanwhile, no significant changes were observed for only *C. ambrosoides* regarding DPPH activity. In accordance with our results, high antioxidant capacities were obtained for some plants, such as tomato plants, when treated with growth promoting bacteria [51]. The endophytes-provoked antioxidant capacities might be exerted as a result of an accumulation of reactive oxygen species by the endophytes or plants. Furthermore, *C. ambrosioides* extracts have previously shown high DPPH scavenging activity and TBARS inhibition [47], while its oil displayed high scavenging antioxidant activity, tested by different assays, including ABTS, DPPH, β-carotene-linoleic acid, and reducing power assays [16,52]. Such activity might be attributed to its rich content of phenolic compounds [47], as well as monoterpene hydrocarbons, particularly α-terpinene [52]. Previous studies have also shown *C. botrys* extracts to display high total antioxidant capacities (evaluated by DPPH, FRAP, and ABTS assays), superoxide anion (O_2_^−^), nitric oxide (NO), and hydroxyl (HO) radicals [50].

### 2.8. Endophytic Bacteria Enhanced the Chenopodium Biological Activities

#### 2.8.1. Antibacterial Activities

The *Chenopodium* species have been known for their promising antimicrobial activities [17]. According to our results, the extracts of all tested species were shown to induce antimicrobial activities against several bacterial and fungal species, whereas the most potent effect was clearly shown by all plants against *Aspergillus flavus*, based on the inhibition zone diameter (Table 6). Interestingly, such potential activities were significantly enhanced in the examined species in response to the *Streptomyces* (strain JSA11) inoculation, while no changes were obtained for some sprouts regarding their activity against certain organisms (e.g., *Enterococcus faecalis* in the case of *C. ficifolium* and *C. botrys*). In accordance, several reports have investigated the potential antimicrobial effects of the *Chenopodium* species. For instance, *C. ambrosioides* oil has previously exhibited potent fungi toxic activity against *Aspergillus flavus* [16]. Meanwhile, *C. ambrosioides* had also antimicrobial activity against *Staphylococcus aureus*, *Micrococcus luteus*, *Bacillus cereus*, *Escherichia coli*, *Pseudomonas aeruginosa* [52], *Bacillus subtilis*, *Aspergillus oryzae,* and *Aspergillus niger* [49]. Moreover, the antifungal activities of *C. ambrosioides* oil on *Candida albicans*, *C. glabrata*, *C. krusei,* and *C. parapsilosis* have been previously investigated [52], the activity that might be attributed to the presence of monoterpene hydrocarbons [52], which could affect the bacterial membranes. Similar to our results, *C. ambrosioides* extracts have been previously revealed to exert an antibacterial effect against *Staphylococcus aureus and Enterococcus faecali**s*. Moreover, *C. ambrosioides* had antimicrobial potency against *Paenibacillus apiarus*, *Paenibacillus thiaminolyticus*, *Mycobacterium tuberculosis*, *M*. *smegmatis,* and *M. avium*. Such activity could be ascribed to the presence of phenolic compounds [53]. On the other hand, *C. botrys* extracts have been shown to possess promising antimicrobial potency against *Staphylococcus aureus* and *Klebsiella pneumonia* [18].

#### 2.8.2. Anti-Inflammatory Activities of Chenopodium Sprouts

In the present investigation, the *Streptomyces* (strain JSA11) bacterial endophytes treatments have resulted in remarkable reductions in COX-2 and LOX activities of both *C. ficifolium* and *C. botrys*, while no significant effect was induced on *C. ambrosoides* when compared with their respective controls (Table 7). Unlike our results, *C. ambrosioides* extracts have been previously shown to exert anti-inflammatory effects against some inflammatory mediators (e.g., K, NO, TNF-α, and PGE2), an effect that might be due to the presence of bioactive compounds, such as phenolics and flavonoids [54]. On the other hand, it has been recently shown that the application of other elicitors, such as eCO_2_ or laser light, has effectively enhanced the anticancer and anti-inflammatory activities of both buckwheat and lemongrass sprouts [3,4].

### 2.9. Species-Specific Response to Endophytic Bacterial Treatment

The hierarchical clustering data represented in (Figure 4) show that there was a clear sprout species-specific response to the effect of the *Streptomyces* (strain JSA11) inoculation. *C. botrys* showed a more prominent response to the enhancing effect of *Streptomyces* bacterial endophytes, where it had the highest content of pigments, vitamins, amino acids, fatty acids, and minerals. Consequently, *C. botrys* showed the highest anti-inflammatory activity, followed by *C. ficifolium.* Meanwhile, *C. ambrosioides* exhibited the highest response to bacterial endophytes regarding the antimicrobial activity. The variations among the three species might be attributed to species diversity and ontogeny [3].

## 3. Material and Methods

### 3.1. Experimental Setup, Plant Materials, and Growth Conditions

#### 3.1.1. Isolation and Characterization of Endophyte Isolates

Endophytic bacteria were isolated from *Chenopodium* leaves and stems, and the isolates were prepared only from *C. ambrosoides.* The dissection of *Chenopodium* leaves was completed, and then they were washed with water and cut into small pieces (1 cm). The dissected pieces were rinsed first with Tween 20 (0.1%) for 30 s), then with Na-hypochlorite (1%) for 5 min, and finally with distilled H_2_O for 5 min. Afterwards, the dissected pieces were surface sterilized in ethanol (70%) for 5 min, and then they were air-dried in a flow chamber. Plant tissue pieces were transferred to plates of a glycerol–yeast–agar medium supplemented with nystatin (50 μg L^−1^). About 1 g of sterilized plant tissue was added to 10 mL of an aqueous saline solution (0.9% NaCl), and then the solution was well mixed and heated at 50 °C for 30 min. Serial dilutions were prepared with the pour plate method. After incubation of 14 days at 27 °C, the selected colonies were purified on the used medium (glycerol–yeast–agar) for 8 days at 27 °C. Further, the purified colonies were maintained on starch casein agar as agar-slants at 4 °C and as suspensions at −20 °C in 20% glycerol until use [55]. The morphological identification was performed by examining cover slips of the isolates with a light microscope. Cultural characteristics in different media were observed after incubation at 27 °C (7–14 days). The bioactive isolates were morphologically characterized using Bergey’s manual key. Biochemical features including IAA and siderophore production were measured [56,57] to investigate the growth promoting potential of endophytic bacteria. In addition, the carbon and nitrogen utilization; hydrolysis of starch, casein and lipid; nitrate reduction, gelatin liquefication, and the ability to produce H_2_S, urease, L-asparaginase, L-glutaminase, and peroxidase were carried out following the methods of [58]. The ability to utilize nitrogen sources was determined in a basal medium containing glucose (2 g), MgSO_4_.7H_2_O (0.125 g), FeSO_4_.7H_2_O (0.002 g), K_2_HPO_4_ (0.25 g), NaCI (0.5 g), and agar (3.0 g) in 200 mL of water.

For antioxidant activities measurement, the purified endophyte strain was cultured on glycerol–yeast extract–agar (glycerol 5 mL, yeast extract 2 g, K_2_HPO_4_ 1 g, agar 15 g, and distilled water 1000 mL) plates at 28 °C for 7 days. The *Streptomyces* strain was inculcated into glycerol–yeast extract broth in Erlenmeyer flasks for propagation. The culture was incubated on a rotary shaker at 180 rpm, at 28 °C, and for 7 days. Then, *Streptomyces* cells were collected by centrifugation for 10 min at 8000× *g* at 4 °C. The precipitated cells were washed in a sterile saline solution, and the cell suspension was freeze-dried and repeatedly extracted with ethanol. After removing the ethanol solvent using a rotary vacuum evaporator at 37 °C, the final extract (30 mg) was obtained and suspended in 10 mL of 80% ethanol (3 mg/mL). Total antioxidant activity, namely, ferric reducing antioxidant power (FRAP), 2-diphenyl-1-picrylhydrazyl (DPPH), and 2,2′-azino-bis (3-ethylbenzothiazoline-6-sulfonic acid) (ABTS) were measured by using an 80% ethanol extract (2 mg/mL). More details about antioxidant measurements are in Section 3.5.1.

The detection of phenolic and flavonoid compounds in the bacterial endophytes was performed via the HPLC method after extraction of 50mg of the cell biomass in methanol according to [3,21]. The compounds were identified by using the standards and their relative retention times. About 25 standards were used and bought from Sigma-Aldrich. The used phenolic and flavonoid standards are commonly present in plants. The peak area of each standard could be used as an indication for the concentrations of each compound. For the detection of the target compounds, about 50 mg of ground sprout samples were mixed with acetone/water (4:1). The HPLC system (SCL-10 AVP, Japan) was provided with a Lichrosorb Si-60, 7 μm, 3 × 150 mm column, and DOD detector. The mobile phase was a mixture of water/formic acid (90:10), as well as acetonitrile/water/formic acid (85:10:5) at a flow rate of 0.8 mL/min. Meanwhile, the internal standard was 3,5-dichloro-4-hydroxybenzoic.

#### 3.1.2. Extraction of DNA

To molecularly identify the bioactive isolate, DNA from the isolates was extracted according to [59]. High quality extracted DNA was used for amplification of the 16S rRNA gene (primers were 1498R (5-ACGGCTACCTTGTTACGACTT-3) and 27F (5-GAGTTTGATCCTGGCTCA-3)) [60,61] Invitrogen (Waltham, MA, USA). The samples were prepared with 50 ng/μL of a PCR product then delivered to the company MacroGen in Korea (http://www.dna.macrogen.com, accessed on 12 May 2021) for nucleotide sequencing.

The evolutionary history was inferred by using the maximum likelihood method and the Tamura 3-parameter model [62]. We generated the tree with the highest log likelihood (-13039.1216). After obtaining the initial tree for the heuristic search (neighbor joining and BioNJ algorithms), a matrix of pairwise distances was estimated (maximum composite likelihood approach), and finally the topology with superior log likelihood value was selected. There was a total of 683 positions in the final dataset. Evolutionary analysis was conducted in MEGA6 [63]. BLAST (http://www.ncbi.nlm.nih.gov/BLAST, accessed on 28 September 2021) was applied for sequence analysis, and then MEGAX software was used for cluster analysis.

#### 3.1.3. Seed Inoculation with Bacterial Endophyte and Growth of Chenopodium Sprouts

The *Chenopodium* seeds (*C. ambrosoides*, *C. ficifolium,* and *C. botrys*) were brought from the Agricultural Research Centre, Giza, Egypt. They were surface sterilized (2.5% sodium hypochlorite, 5 min) and rinsed thoroughly in sterile distilled water. Afterwards, sterilized *Chenopodium* seeds were soaked in a liquid suspension of inoculants at 25% concentrations (2.5 × 10^−7^ cfu.mL^−1^) for 6 h at room temperature, and the control was soaked in distilled water. The control and treated *Chenopodium* seeds were transferred to the sterilized trays filled with vermiculite and irrigated every day. The size of each tray was (8 cm × 12 cm × 4 cm), where sprouts of each treatment were grown in different trays. The *Chenopodium* sprouts were cultured in growth chambers at 24 °C, photosynthetically active radiation (PAR) of 200 µmol m^−2^ s^−1^, 16 h light/8 h dark cycle, and 60% relative humidity. After eight days of cultivation, 30 sprouts from each tray of 8 × 12 × 4 cm in size (a biological replicate) were harvested and weighed to determine the fresh sprout weight. Finally, the sprouts were frozen in liquid nitrogen at −196 °C and stored for further analyses. At least four biological replicates were used for evaluation of chemical and biological attributes. The percentage of water content of the sprouts was estimated by drying fresh sprouts in an oven at 70 °C for 3 days. After determination of the dried weight of the sprouts, the water content was measured as a percentage. The fresh sprout weight (FW) was expressed as biomass (g/sprout).

### 3.2. Determination of Photosynthetic Rate

An EGM-4 infrared gas analyzer (PP Systems, Hitchin, UK) was used for determination of photosynthetic rate, from 180 s measurements of net CO_2_ exchange (NE).

### 3.3. Pigment Analysis

The homogenization of sprouts (about 200 mg) was carried out in acetone (7000 rpm, 1 min) by using a MagNALyser (Roche, Vilvoorde, Belgium). Then they were centrifuged (14,000× *g*, 4 °C, 20 min). The supernatant was filtered and analyzed by using HPLC (Shimadzu SIL10-ADvp, reversed-phase, at 4 °C) [3]. Carotenoids were separated on a silica-based C18 column, and two solvents were used: acetonitrile/methanol/water (81:9:10) and methanol/ethyl acetate (68:32). In addition, chlorophyll a and b and β-carotene were extracted and detected at 4 wavelengths (420, 440, 462, and 660 nm) by using a diode-array detector (Shimadzu SPDM10Avp).

### 3.4. Determination of the Nutritional Value

In order to give insight into the functional food value of *Chenopodium* sprouts, the levels of amino, organic, and fatty acids, as well as minerals and vitamins profiles were evaluated as described below.

#### 3.4.1. Proximate Composition Analysis

Carbohydrate levels were evaluated following the method of [64] from each *Chenopodium* sprout treated and nontreated with bacterial endophytes. The protein concentration was also measured for each sprout sample (0.2 g FW) [65]. Total lipid concentrations were assessed, where the sprout samples were homogenized in a 2:1 mixture of chloroform/methanol (*v*/*v*) [66]. Then sprouts were centrifuged for 15 min at 3000× *g*. The pellets were redissolved in a 4:1 mixture of toluene/ethanol (*v*/*v*). After concentration, the total lipid content was calculated. The extracted lipids were determined by gravimetric analysis and expressed as weight (g) per fresh weight (g) of sprout. Crude fibers were extracted, where sprouts were gelatinized (heat-stable alpha-amylase, pH 6, 100 °C for 25 min), and then enzymatically digested (protease: pH 7.5, 60 °C, 25 min and amyloglucosidase: pH 6, 0 °C, 30 min) to remove undesirable protein and starch [67]. Fibers were precipitated in ethanol for washing, and then after washing, the residues were weighed.

#### 3.4.2. Elemental Analysis

The detection of mineral elements was performed according to [68], whereas 200 mg from bacterial endophytic-treated and control sprouts were digested in 5:1 (*v*:*v*) HNO_3_/H_2_O solution. Thereafter, macro and microelements were evaluated (inductively coupled plasma mass spectrometry, ICP-MS, Finnigan Element XR, and Scientific, Bremen, Germany). Nitric acid (1%) was used as blank.

#### 3.4.3. Amino Acids Levels and Metabolism

The method described in [69] was used, whereas 100 mg of each plant was homogenized in 5 mL of 80% ethanol at 5000 rpm for 1 min. After centrifugation (14,000× *g* for 25 min), the supernatant was resuspended in 5 mL chloroform. Thereafter, 1 mL of H_2_O was used for the residue extraction. The supernatant and pellet were resuspended in chloroform and centrifuged (8000× *g*, 10 min). Finally, the amino acids were quantified (Waters Acquity UPLC TQD device coupled to a BEH amide column), the elution (A, 84% ammonium formate, 6% formic acid, and 10% acetonitrile, *v*/*v*), and (B, acetonitrile and 2% formic acid, *v*/*v*).

Following the protocol of [4], glutamine synthase (GS) activity was determined, and the extraction was performed in (100 mg mL^−1^ Tris-HCl (50 mM), pH 7.4, 2% polyvinylpyrrolidone, 4 mM DTT, MgC1_2_ (10 mM), EDTA (1 mM), 10% glycerol, and 2 mM PMSF). Then, the GS activity was evaluated in a Tris-acetate reaction buffer (Tris-acetate, 200 mM, pH 6.4), as evidenced by the production of γ-glutamyl hydroxamate. Dihydrodipicolinate synthase (DHDPS) activity was performed according to [70]. Tested sprouts without L-aspartate-b-semialdehyde were used as negative controls. The reaction was performed at 36.5 °C to allow the adduct formation between the reaction product and o-ABA. The reaction was stopped by addition of 12% of trichloroacetic acid (TCA). After dark, incubation for 60 min, samples were measured at 550 nm. Cystathionine γ- synthase (CGS) was extracted in 20 mM MOPS for 15 min at 4 °C, and the supernatants were mixed with a reaction buffer containing L-cysteine (2 mM), PLP (100 μM), AVG (200 μM), and O-phospho-homoserine (5 mM). L-cystathionine formation was separated on a phenomenex Hyperclone C18 BDS column (Dionex HPLC system) [71].

#### 3.4.4. Organic Acid Analysis

The detection of organic acids was conducted in sprouts samples (200 mg) by using HPLC (0.001 N sulfuric acid, at 210 nm, 0.6 mL min^−1^) [72]. A liquid chromatographer (Dionex, Sunnyvale, CA, USA), and a LED model detector (Ultimate 3000) were used for detection. In cooperation, an EWPS-3000SI autosampler, a TCC-3000SD column thermostat, and an LPG-3400A pump were included into the system. Chromeleon v.6.8 computer software was also applied. Then, separation was performed on an Aminex HPH-87 H (300 × 7.8 mm) column with IG Cation H (30 × 4.6) precolumn of Bio-Red firm at a temperature of 65 °C.

#### 3.4.5. Fatty Acids Analysis

The detection of fatty acids using GC/MS (Hewlett Packard, Palo Alto, CA, USA) equipped with an HP-5 MS column (30 m × 0.25 mm × 0.25 mm), where 200 mg were taken from the sprout samples for extraction [73]. The database NIST 05 and Golm Metabolome were applied (http://gmd.mpimp-golm.mpg.de, accessed on 8 December 2021).

#### 3.4.6. Vitamin Level Analysis

The contents of thiamine and riboflavin were determined in sprouts (about 200 mg fresh samples) by using UV and/or fluorescence detectors [3]. Separation was performed on a reverse-phase (C18) column (HPLC, methanol/water). Vitamin C (ascorbate) was determined by HPLC analysis (Shimadzu, Hertogenbosch, The Netherlands). Sprouts tissues were extracted in 1 mL of ice-cold 6% (*w*/*v*) meta-phosphoric acid, and antioxidants were separated on a reversed phase HPLC column [74]. Vitamin E (tocopherols) was extracted with hexane. The dried extract was resuspended in hexane, and tocopherols were separated and quantified by HPLC (Shimadzu, Hertogenbosch, The Netherlands) (normal phase conditions, Particil Pac 5 µm column material, length 250 mm, i.d. 4.6 mm) [74].

#### 3.4.7. Determination of Phenolics Levels and Their Biosynthetic Enzyme Activity

The total phenolic and flavonoid contents were evaluated by homogenizing (120 mg of sprouts in 80% ethanol). Centrifugation was performed at (4 °C, 20 min), and then the phenolic content was determined by using a Folin–Ciocalteu assay, where gallic acid was used as a standard. The flavonoid content was evaluated following the modified aluminum chloride colorimetric method, where quercetin was applied as a standard [3].

Phenylalanine ammonia-lyase (PAL) was extracted from 0.25 g (FW) frozen plant material in 1 mL sodium borate buffer (200 mM, pH 8.8) and assayed in a Tris-HCl (100 mM, pH 8.8) reaction buffer containing L-phenylalanine (40 mM) by measuring the absorbance of the produced transcinnamic acid at 290 nm [68]. Samples sprouts were replaced with water to serve as a negative control.

#### 3.4.8. Phosphate Solubilization

The phosphate (PO_4_) solubilization was measured by incubating the samples into a medium (100 mL) containing tri-Ca-PO_4_ for a week at 30 °C. The pH was recorded (pH meter, Germany). The produced soluble PO_4_ was spectrophotometrically measured [75].

### 3.5. Biological Activities

#### 3.5.1. Antioxidant Activities

The antioxidant capacities of the sprouts were evaluated by using different assays [3]. For determination of ferric reducing antioxidant power (FRAP), about 0.1 g was extracted in 80% ethanol. Then, centrifugation took place (14,000 rpm, 20 min). Afterwards, about 0.1 mL extract was mixed with FRAP reagent (20 mM FeCl_3_ in 0.25 M acetate buffer). For determination of the 2,2′-azino-bis(3-ethylbenzothiazoline-6-sulfonic acid) (ABTS), the ABTS was mixed with 2.4 mM potassium persulphate. The absorbance was detected at 734 nm, while detection of DPPH activity was performed by using 0.1 mL of the extract and 0.25 mL of the DPPH reagent. The absorbance was detected at 517 nm.

#### 3.5.2. Antibacterial Activity

The standard dilution method was used for measuring the antibacterial activity of the sprouts [4]. The sprout extract (100 mg) was mixed with 1 mL dimethylosulfoxide (DMSO). Then, 0.1 mL of liquid culture of standard strain (*Staphylococcus aureus* ATCC 6538 P) was diluted 1:10.000 in the same medium (number of inoculums contained 104–105 bacterial cells in 1 mL) and was added to the media. Afterwards, the samples were incubated at 37 °C for 18 h. The MIC (Minimal Inhibitory Concentration) of the tested sprouts was detected, and the antibacterial activity was tested against *Pseudomonas aeruginosa* (ATCC10145)*, Candida glabrata* (ATCC90030)*, Proteus vulgaris* (ATCC8427)*, Enterobacter aerogenes* (ATCC 13048)*, Staphylococcus saprophyticus* (ATCC 19701)*, E. coli* (ATCC 29998)*, Salmonella typhimurium* (ATCC14028), *S. epidermidis* (ATCC 12228)*, Candida albicans* (ATCC90028)*, Salmonella typhimurium* (ATCC14028)*, Enterococcus faecalis* (ATCC 10541)*, Streptococcus salivarius* (ATCC25975)*, Aspergillus flavus* (ATCC9170), and *Serratia marcescens* (ATCC99006). Antibiotics, such as ciprofloxacin at 25 mg/mL and 100% DMSO, were used as positive and negative controls, respectively.

#### 3.5.3. Determination of Lipoxygenase (LOX) and Cyclooxygenase (COX) Activities

Sprouts tissue (1.5 g) from each of the *Chenopodium* species investigated was extracted with 80% ethanol (10 mL). After shaking for 60 min, samples were centrifuged at 2500 rpm for 15 min, and the supernatants were filtered (Whatman No.1). After drying, the extract was reconstituted in 100% dimethyl sulphoxide (DMSO) (Merck Schuchardt OHG) at 10 mg/mL and tested in the assays.

Lipoxygenase activity was evaluated by using linoleic acid as a substrate and LOX as an enzyme [4]. About 10 μL of the sprout extract was mixed with 90 μL LOX (400 U/mL). Then, the mixture was incubated at 25 °C for 5 min. Afterwards, 100 μL of linoleic acid (0.4 mM) was added, and then incubation was performed again for 20 min at 25 °C. Thereafter, about 100 μL of ferrous orange xylenol reagent in 10 μM FeSO_4_, 30 mM H_2_SO_4_, 90% methanol, and 100 μM xylenol orange were added. The absorbance was detected at 560 nm, and the percentage of inhibition was evaluated.

Cyclooxygenase-2 activity inhibition was detected by applying the manufacturer’s instructions for the COX assay kit (No. 560131; Cayman Chemical, Ann Arbor, MI, USA). Incubation was performed for 90 min at 25 °C. The absorbance was detected at 420 nm, and the percentage of inhibition was measured [4].

### 3.6. Statistical Analyses

Statistical analyses were determined by using the SPSS package (SPSS Inc., Chicago, IL, USA). A one-way analysis of variance (ANOVA) was used for all data per species. Tukey’s multiple range test (*p* < 0.05) was performed as the post hoc test for mean separations. Each experiment was performed in three replicates (*n* = 3). All parameters were subjected to Pearson’s distance metric cluster analysis by using the MultiExperiment Viewer (MeV) TM4 software package (Dana-Farber Cancer Institute, Boston, MA, USA).

## 4. Conclusions

Based on our obtained results, a selected bioactive endophyte *Streptomyces* (strain JSA11) could represent a promising approach to enhance the growth and yield of the tested *Chenopodium* sprouts and to increase their contents of bioactive metabolites, such as amino acids, organic acids, vitamins, and minerals. At the ecological level, endophytic bacteria could represent an ecofriendly approach and possibly an alternative to chemical fertilizers, with efficient plant growth promoting activity. Meanwhile, at the economic level, endophytic bacterial treatment and also sprouting are considered low-cost techniques, particularly when combined together to ensure higher plant productivity as well as increased levels of phytochemicals and biological activities.

## Figures and Tables

**Figure 1 plants-10-02745-f001:**
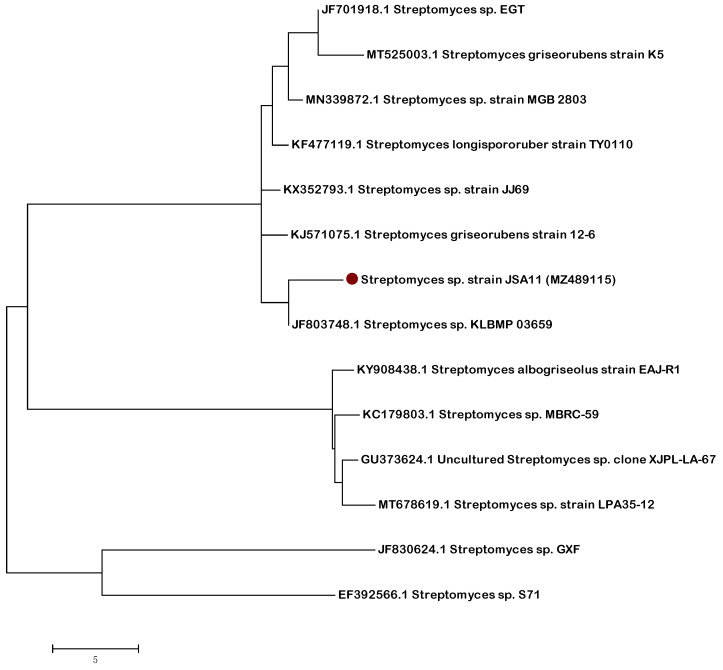
Neighbor joining tree (partial sequences ~ 950 bp) showing the phylogenetic relationships of actinobacterial 16S rRNA gene sequence of potential strains to closely related (S ≥ 97%) sequences from the GenBank database.

**Figure 2 plants-10-02745-f002:**
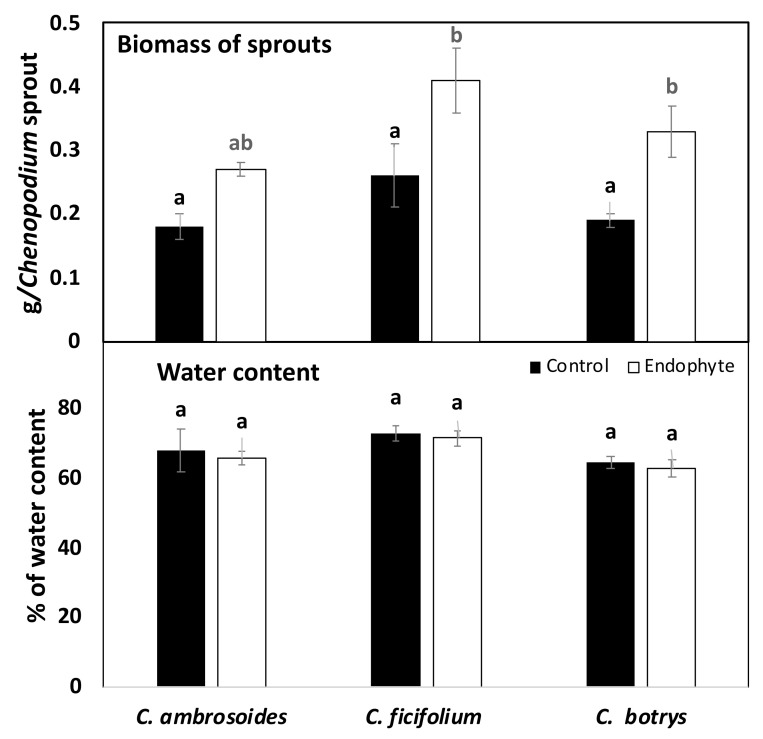
Biomass and water content of control and endophytic bacterial-treated *Chenopodium ambrosoides, Chenopodium ficifolium,* and *Chenopodium botrys* sprouts. Data are represented by the means of at least 3 replicates, and error bars represent standard deviations. Different small letters above bars indicate significant differences between means at *p* < 0.05.

**Figure 3 plants-10-02745-f003:**
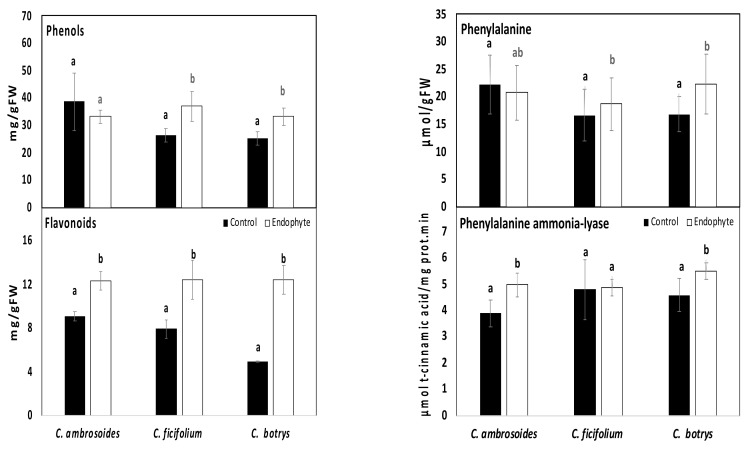
Phenolic, flavonoids, and phenolic biosynthesis enzymes of control and endophytic bacterial-treated *Chenopodium ambrosoides, Chenopodium ficifolium,* and *Chenopodium botrys* sprouts. Data are represented by the means of at least 3 replicates, and error bars represent standard deviations. Different small letters above bars indicate significant differences between means at *p* < 0.05.

**Figure 4 plants-10-02745-f004:**
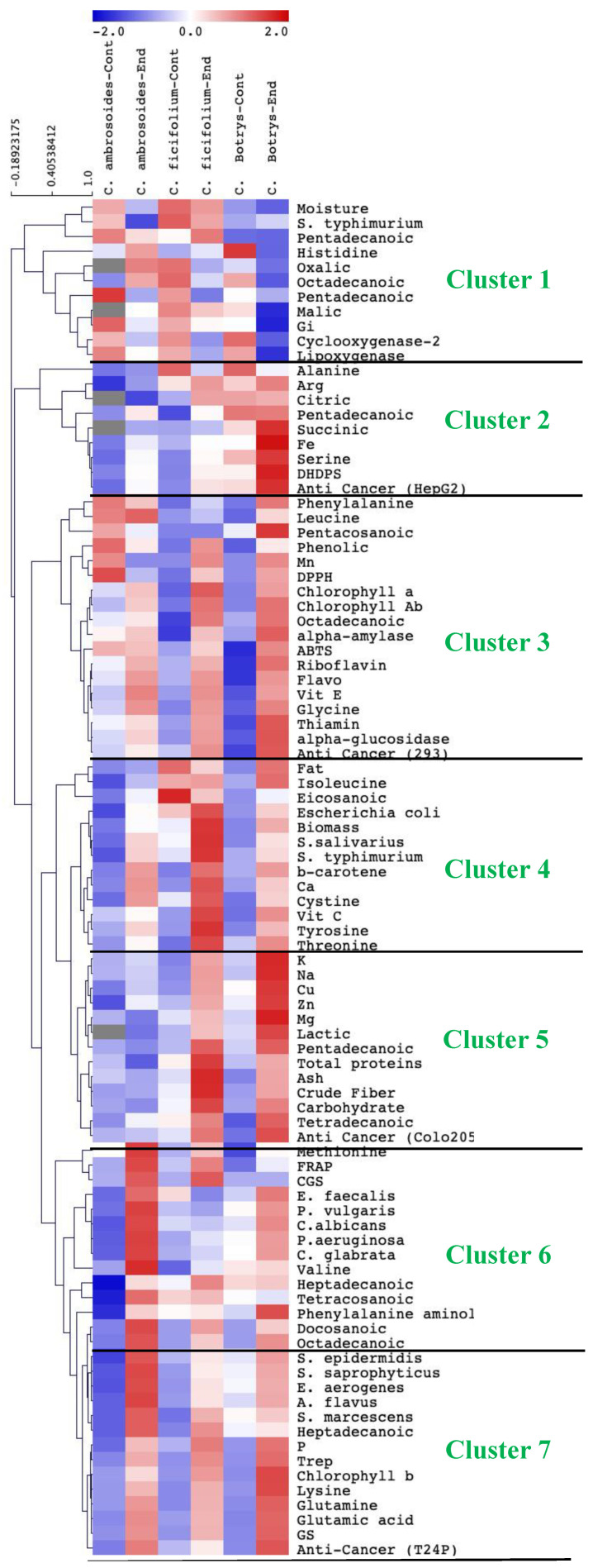
Species-specific responses of *Chenopodium ambrosoides, Chenopodium ficifolium,* and *Chenopodium botrys* sprouts to the effect of endophytic bacterial treatment on the nutritional and health-promoting properties. The measured parameters are represented by amino, fatty, and organic acids; antioxidant capacity; the contents of pigments, minerals, and vitamins; and antibacterial, antidiabetic, anti-inflammatory, and anticancer activities. Data are represented by the means of at least 3 replicates.

**Table 1 plants-10-02745-t001:** The morphological and biochemical characterization of bacterial isolates, whereas the signs + and − indicate presence or absence, respectively. Data are represented by the means of at least 3 replicates, and error bars represent standard deviations.

	Isolate	Endo 1	Endo 2	Endo 3	Endo 4	Endo 5	Endo 6
Colony	Aerial mycelium	+	−	+	+	−	+
	Pigmentation	+	+	+	+	+	+
Spore chain	Spiral	+	−	−	+	+	−
	Rectiflexibles	−	+	+	−	−	−
	Verticillate	−	−	−	−	−	+
Spore color	Yellow	+	−	−	−	−	−
	Orange	−	+	+	+	−	+
	Red	−	−	−	−	+	−
N source utilization	L-Cysteine	+	+	+	+	+	−
	L-Phenylalanine	−	−	+	−	−	
	L-Histidine	−	−	+	+	+	−
	L-Lysine	+	+	+	−	−	+
	L-Asparagine	+	-	+	+	+	+
	L-Arginine	+		−	−	−	−
	L-proline	+	+	+	+	−	
	L-Valine	−	−	+	−	−	+
	Tyrosine	+	+	−	+		+
C source utilization	D-fructose	−	−	+	+	+	−
	D-glucose	+	+	−	+	−	−
	Sucrose	+	+	+	-	+	+
	Maltose	−	−	+	−	+	
	Raffinose	+	+	−	+	+	+
	Lactose	−	−		+	+	−
	Galactose	+	+	−	+	+	−
	Meso-Inositol	+	+	+	−	+	+
	Cellulose	−	−	+	+	+	+
	Xylose	+	+	+	−	+	−
	Dextran	+	+	−	−	−	−
Enzyme activity	Catalase	+	−	+	−	+	−
	Peroxidase	+	−	+	+	+	+
	Starch hydrolysis	+		+	−	−	+
	Gelatin liquefication	+	+	−	+	−	+
	Casein hydrolysis	−	−	+	+	+	+
	Lipolysis	+	+	+	+	−	+
	Citrate utilization	+	+	+	+	+	+
	H_2_S Production	−	+	−	+	+	+
	Nitrate reduction	+	+	+	−	−	−
	Urease	+	+	+		−	−
	L-Asparaginase	−	+	+	+	+	−
	L-Glutaminase	+	+	+	+	+	+
Biological activity	Antioxidant Activity (FRAP)	42.4 ± 3.1	66.6 ± 4.0	21.8 ± 1.6	33.1 ± 1.5	24 ± 1.4	32.4 ± 2.3
	Antioxidant Activity DPPH (%)	68.7 ± 2.8	73.7 ± 3.8	34.2 ± 2.1	20.8 ± 1.5	27.5 ± 1.4	48.2 ± 2.9
	Antioxidant Activity (ABTS%)	30.5 ± 1.0	59.6 ± 1.9	39.7 ± 1.3	34.8 ± 1.1	23.2 ± 0.8	53.6 ± 1.7
	Phosphate Solubilization (mg/mL)	5.7 ± 0.8	7.2 ± 0.2	4.5 ± 0.1	3.8 ± 1.5	2.7 ± 1.4	7.3 ± 1.0
Bioactive compounds production	Total flavonoids (mg/100 g bacteria)	5 ± 1.19	8 ± 1.53	5.8 ± 0.84	5.3 ± 1.36	6.4 ± 0.86	7.5 ± 1.18
	Total Phenols (mg/100 g bacteria)	36.8 ± 1.06	45.4 ± 1.31	23.7 ± 0.6	42.2 ± 1.2	23.7 ± 0.6	33.8 ± 0.9
	Tocopherols (mg/g bacteria)	0.3 ± 0.01	0.5 ± 0.01	0.2 ± 0.01	0.5 ± 0.01	0.3 ± 0.01	0.2 ± 0.01
	Flavonoids (mg/100 g bacteria)						
	Quercetin		1.63 ± 0.1				
	Quercetrin		1.41 ± 0.5				
	Luteolin		0.70 ± 0.1				
	Apigenin		4.11 ± 0.7				
	Isoquercetrin		10.2 ± 1.6				
	Rutin		1.27 ± 0.4				
	Ellagic acid		0.71 ± 0.1				
	Velutin		0.30 ± 0.0				
	Naringenin		1.12 ± 0.3				
	Genistein		0.95 ± 0.2				
	Daidzein		1.08 ± 0.1				
	Fisetin		0.74 ± 0.1				
	O-hydroxydaidzein		1.13 ± 0.1				
	IAA-Me		1.15 ± 0.21				
	ABA		0.29 ± 0.1				
	GA		0.16 ± 0.07				
	Sidephore Catechol		7.3 ± 0.3				
	Sidephore Salicylate		9.23 ± 0.42				

**Table 2 plants-10-02745-t002:** Photosynthetic-related parameters and pigment contents of control and endophytic bacterial-treated *Chenopodium ambrosoides, Chenopodium ficifolium,* and *Chenopodium botrys* sprouts. Data are represented by the means of at least 3 replicates ± standard deviations. Different small letter superscripts (a, b) within a row indicate significant differences between control and endophytic bacterial samples.

	*C. ambrosoides*		*C. ficifolium*		*C. botrys*	
	Control	Endo	Control	Endo	Control	Endo
**Photosynthetic Related Parameters (** **μmol CO_2_ m^−2^ s^−1^)**						
Photosynthesis	10.1 ± 0.8 ^a^	11.5 ± 0.8 ^b^	11.3 ± 1.1 ^a^	13.1 ± 0.7 ^b^	9.7 ± 1 ^a^	12.8 ± 1.2 ^b^
Respiration	1.4 ± 5.4 ^a^	1.9 ± 0.0 ^ab^	1.2 ± 0.06 ^a^	2.0 ± 0.1 ^b^	1.1 ± 0.0 ^a^	1.9 ± 0.1 ^b^
**Pigments (mg/gFW)**						
Chl ^a^	2.07 ± 0.4 ^a^	2.42 ± 0.38 ^a^	1.65 ± 0.2 ^a^	2.78 ± 0.4 ^b^	1.76 ± 0.3 ^a^	2.55 ± 0.4 ^b^
Chl ^b^	1.04 ± 0.1 ^a^	1.46 ± 0.2 ^a^	1.02 ± 0.1 ^a^	1.65 ± 0.3 ^ab^	1 ± 0.05 ^a^	1.89 ± 0.41 ^b^
Chl ^a+b^	3.11 ± 0.5 ^a^	3.8 ± 0.3 ^a^	2.68 ± 0.4 ^a^	4.4 ± 0.41 ^a^	2.7 ± 0.3 ^a^	4.44 ± 0.8 ^b^
Beta-carotene	0.06 ± 0.01 ^a^	0.11 ± 0.02 ^b^	0.06 ± 0.0 ^a^	0.12 ± 0.02 ^b^	0.07 ± 0.01 ^a^	0.11 ± 0.01 ^ab^

**Table 3 plants-10-02745-t003:** Proximate composition, vitamins, minerals of control, and endophytic bacterial-treated *Chenopodium ambrosoides, Chenopodium ficifolium,* and *Chenopodium botrys* sprouts. Data are represented by the means of at least 3 replicates ± standard deviations. Different small letter superscripts (a, b) within a row indicate significant differences between control and endophytic bacterial samples.

	*C. ambrosoides*		*C. ficifolium*		*C. botrys*	
	Control	Endo	Control	Endo	Control	Endo
**Minerals (mg/gDW)**						
K	13.9 ± 0.4 ^a^	15.97 ± 1.2 ^ab^	11.4 ± 0.12 ^a^	23.2 ± 1 ^b^	15.5 ± 1 ^a^	29.8 ± 1.3 ^b^
Na	1.8 ± 0.04 ^a^	2.1 ± 0.3 ^b^	1.5 ± 0.01 ^a^	3.2 ± 0.1 ^b^	1.9 ± 0.1 ^a^	4.1 ± 0.1 ^b^
Ca	1.46 ± 0.3 ^a^	2.1 ± 0.12 ^b^	1.4 ± 0.17 ^a^	2.28 ± 0.1 ^b^	1.5 ± 0.2 ^a^	1.95 ± 0.02 ^a^
Cu	0.006 ± 0 ^a^	0.008 ± 0 ^ab^	0.007 ± 0 ^a^	0.011 ± 0 ^b^	0.01 ± 0 ^a^	0.013 ± 0 ^a^
Fe	0.13 ± 0.02 ^a^	0.19 ± 0 ^b^	0.16 ± 0.01 ^a^	0.2 ± 0.02 ^a^	0.2 ± 0.01 ^a^	0.33 ± 0.04 ^ab^
P	0.96 ± 0.1 ^a^	1.4 ± 0.05 ^ab^	1.13 ± 0.07 ^a^	1.62 ± 0.05 ^b^	1.06 ± 0.0 ^a^	1.64 ± 0.2 ^a^
Zn	0.09 ± 0 ^a^	0.12 ± 0.01 ^ab^	0.11 ± 0 ^a^	0.14 ± 0.01 ^a^	0.12 ± 0.0 ^a^	0.16 ± 0.01 ^a^
Mn	0.028 ± 0 ^a^	0.024 ± 0 ^a^	0.027 ± 0 ^a^	0.03 ± 0 ^a^	0.02 ± 0 ^a^	0.031 ± 0 ^ab^
Mg	2.8 ± 0.3 ^a^	2.3 ± 0.2 ^a^	3.18 ± 0.39 ^a^	3.9 ± 0.2 ^a^	3.1 ± 0.5 ^a^	5.18 ± 0.1 ^b^
**Vitamins (mg/gFW)**						
Vitamin C	2.09 ± 0.07 ^a^	2.34 ± 0.42 ^b^	1.92 ± 0.06 ^a^	3.04 ± 0.61 ^b^	1.7 ± 0.72 ^a^	2.7 ± 0.8 ^b^
Vitamin E	4.71 ± 0.2 ^a^	6.69 ± 0.23 ^b^	4.25 ± 0.42 ^a^	6.54 ± 0.58 ^b^	3.46 ± 0.2 ^a^	6.4 ± 0.6 ^b^
Thiamin	0.49 ± 0.1 ^a^	0.53 ± 0.14 ^a^	0.42 ± 0.05 ^a^	0.58 ± 0.13 ^a^	0.35 ± 0.07 ^a^	0.64 ± 0.1 ^b^
Riboflavin	0.15 ± 0.03 ^a^	0.18 ± 0.05 ^a^	0.13 ± 0.03 ^a^	0.18 ± 0.02 ^a b^	0.09 ± 0.01 ^a^	0.2 ± 0.04 ^b^
**Proximate composition (mg/gFW)**						
Total proteins	9.1 ± 1 ^a^	6.5 ± 4.8 ^b^	11.2 ± 1.8 ^a^	16 ± 3.7 ^b^	9.1 ± 1 ^a^	13.1 ± 3.2 ^b^
Fat	123 ± 5.4 ^a^	125 ± 15 ^a^	151.3 ± 6 ^a^	140. ± 6 ^b^	123 ± 5.4 ^a^	150.1 ± 16 ^b^
Crude Fiber	6.6 ± 0.6 ^a^	6.8 ± 1.8 ^a^	8.1 ± 0.7 ^a^	12.3 ± 1 ^b^	6.6 ± 0.6 ^a^	10.1 ± 0.8 ^b^
Ash	3.8 ± 0.5 ^a^	3.6 ± 0.4 ^a^	3.9 ± 0.5 ^a^	5.3 ± 0.8 ^b^	3.4 ± 0.5 ^a^	4.6 ± 0.7 ^b^
Carbohydrate	6.3 ± 0.3 ^a^	5.9 ± 0.2 ^a^	7.7 ± 0.4 ^a^	11.1 ± 0.8 ^b^	6.3 ± 0.3 ^a^	10.1 ± 2 ^b^

**Table 4 plants-10-02745-t004:** Amino acids and amino acids-related enzymes, organic acids, and fatty acids of control and endophytic bacterial -treated *Chenopodium ambrosoides, Chenopodium ficifolium,* and *Chenopodium botrys* sprouts. Data are represented by the means of at least 3 replicates ± standard deviations. Different small letter superscripts (a, b) within a row indicate significant differences between control and endophytic bacterial samples.

	*C. ambrosoides*		*C. ficifolium*		*C. botrys*	
	Control	Endo	Control	Endo	Control	Endo
**Amino Acid Metabolism (** **µg/gFW)**						
Glutamic acid	14.6 ± 1.1 ^a^	19.2 ± 1.8 ^b^	14.98 ± 1 ^a^	18.4 ± 2.7 ^b^	14.5 ± 1.4 ^a^	19.9 ± 1.3 ^b^
Glutamine	12.8 ± 0.9 ^a^	16.4 ± 1.1 ^b^	12.5 ± 0.9 ^a^	16.0 ± 1.2 ^b^	12.6 ± 1.3 ^a^	17.4 ± 1.3 ^b^
Serine	6.39 ± 0.3 ^a^	8.61 ± 1.1 ^b^	6.7 ± 0.5 ^a^	8.8 ± 0.8 ^b^	9.8 ± 2.4 ^a^	11.8 ± 1.9 ^c^
Glycine	8.3 ± 0.13 ^a^	9.6 ± 0.0 ^b^	7.6 ± 0.12 ^a^	9.5 ± 0.7 ^b^	7.3 ± 0.66 ^a^	9.7 ± 0.18 ^b^
Arg	13.6 ± 0.5 ^a^	16 ± 1.5 ^ab^	20.9 ± 3.5 ^a^	23.3 ± 2.3 ^b^	21.6 ± 3.8 ^a^	24.2 ± 1.8 ^b^
Alanine	2.4 ± 0.3 ^a^	2.48 ± 0.8 ^a^	3.5 ± 0.16 ^a^	2.7 ± 0.5 ^b^	3.4 ± 0.8 ^a^	2.85 ± 0.5 ^a^
Histidine	5.2 ± 0.48 ^a^	5.7 ± 0.2 ^ab^	4.98 ± 0.8 ^a^	5.2 ± 0.29 ^a^	6.1 ± 0.42 ^a^	4.6 ± 0.7 ^b^
Valine	5.3 ± 0.7 ^a^	6.95 ± 1 ^b^	5.05 ± 1.3 ^a^	5.72 ± 0.9 ^a^	5.9 ± 0.8 ^a^	6.0 ± 1.04 ^a^
Methionine	2.2 ± 0.19 ^a^	2.9 ± 0.3 ^ab^	2.01 ± 0.2 ^a^	3.5 ± 0.4 ^b^	1.65 ± 0.1 ^a^	3.31 ± 0.0 ^b^
Cystine	1.4 ± 0.03 ^a^	2.1 ± 0.18 ^b^	1.7 ± 0.06 ^a^	2.3 ± 0.2 ^b^	1.4 ± 0.12 ^a^	1.9 ± 0.1 ^ab^
Isoleucine	4.2 ± 0.2 ^a^	5.48 ± 0.3 ^a^	7.15 ± 1 ^a^	7.3 ± 1.5 ^a^	5.2 ± 0.79 ^a^	7.8 ± 0.9 ^b^
Leucine	13.8 ± 0.4 ^a^	14 ± 0.9 ^ab^	11.5 ± 1.4 ^a^	12.0 ± 1.1 ^a^	11.1 ± 0.7 ^a^	13.0 ± 1 ^ab^
Tyrosine	5.6 ± 0.7 ^a^	6.3 ± 0.5 ^ab^	5.47 ± 0.4 ^a^	7.2 ± 0.2 ^b^	5.3 ± 0.1 ^a^	6.4 ± 0.6 ^ab^
Lysine	13.5 ± 0.3 ^a^	21.1 ± 1.6 ^b^	12.8 ± 0.6 ^a^	21.3 ± 1.3 ^b^	13.3 ± 0.7 ^a^	25.8 ± 2.5 ^b^
Threonine	4.6 ± 0.3 ^a^	5.3 ± 0.6 ^ab^	4.4 ± 0.4 ^a^	6.3 ± 0.2 ^b^	4.9 ± 0.1 ^a^	5.9 ± 0.2 ^ab^
Trep	0.4 ± 0.04 ^a^	0.61 ± 0.01	0.4 ± 0.04 ^a^	0.6 ± 0.03	0.4 ± 0.04 ^a^	0.67 ± 0.04
**Amino Acid Biosynthesis Enzymes**						
GS (nmol γ-glutamyl hydroxamate/mg protein min^−1^)	5.4 ± 0.4 ^a^	7.14 ± 0.5 ^b^	5.51 ± 0.4 ^a^	6.8 ± 0.7 ^ab^	5.44 ± 0.5 ^a^	7.4 ± 0.7 ^b^
DHDPS (nmol o-ABA and L-2,3- dihydrodipicolinate adduct/mg protein min^−1^)	2.09 ± 0.3 ^a^	2.4 ± 0.2 ^a^	2.09 ± 0.3 ^a^	3.5 ± 0.2 ^b^	2.5 ± 0.4 ^a^	3.1 ± 0.3 ^b^
CGS (nmol L-cystathionine/mg protein min^−1^)	0.012 ± 0 ^a^	0.033 ± 0 ^b^	0.013 ± 0 ^a^	0.039 ± 0 ^b^	0.014 ± 0 ^a^	0.37 ± 0 ^b^
**Organic Acid (mg/g FW)**						
Oxalic	9.08 ± 0.8 ^a^	9.5 ± 0.4 ^a^	3.5 ± 0.5 ^a^	4.73 ± 4.7 ^b^	1.92 ± 0.3 ^a^	4.73 ± 0.5 ^b^
Malic	2.3 ± 0.31 ^a^	3.58 ± 0.2 ^ab^	2.92 ± 0.3 ^a^	2.67 ± 0.6 ^a^	2.1 ± 0.22 ^a^	4.27 ± 0.9 ^b^
Succinic	2.17 ± 0 ^a^	2.03 ± 0 ^a^	1.75 ± 0.3 ^a^	3.5 ± 0.3 ^a^	3.2 ± 1.2 ^a^	5.9 ± 0.9 ^b^
Citric	1.3 ± 0.04 ^a^	1.9 ± 0.09 ^a^	2.96 ± 0.29 ^a^	2.9 ± 0.2 ^a^	2.88 ± 0.5 ^a^	2.5 ± 0.5 ^ab^
Lactic	0.2 ± 0.1 ^a^	0.24 ± 0.1 ^a^	0.32 ± 0.09 ^a^	0.26 ± 0.07 ^a^	0.33 ± 0.05 ^a^	0.49 ± 0.05 ^ab^
**Fatty Acids (mg/g FW)**						
Tetradecanoic (C14:0)	0.7 ± 0.1 ^a^	0.8 ± 0.1 ^ab^	0.8 ± 0.1 ^a^	0.98 ± 0.09 ^b^	0.66 ± 0.06 ^a^	1.02 ± 0.1 ^b^
Pentadecanoic (C16:0)	12 ± 2 ^a^	15.8 ± 1 ^ab^	11.6 ± 0.6 ^a^	15.5 ± 0.7 ^ab^	18 ± 1.6 ^a^	18.1 ± 1.3 ^a^
Eicosanoic (C20:0)	0.8 ± 0.03 ^a^	0.96 ± 0.1 ^ab^	1.1 ± 0.2 ^a^	1.02 ± 0.1 ^a^	0.88 ± 0.09 ^a^	0.96 ± 0.1 ^a^
Docosanoic (C22:0)	0.8 ± 0.1 ^a^	1.45 ± 0.2 ^b^	0.91 ± 0.1 ^a^	1.29 ± 0.1 ^ab^	0.91 ± 0.1 ^a^	1.2 ± 0.1 ^ab^
Octadecanoic (C18:0)	6.9 ± 0.7 ^a^	10.2 ± 1.8 ^b^	7.2 ± 0.8 ^a^	8.9 ± 0.9 ^ab^	7.2 ± 1.2 ^a^	9.5 ± 1.2 ^b^
Pentacosanoic (C24:0)	0.14 ± 0.0 ^a^	0.13 ± 0.02 ^a^	0.12 ± 0.01 ^a^	0.12 ± 0.01 ^a^	0.13 ± 0.01 ^a^	0.15 ± 0.1 ^a^
Pentadecanoic (C16:1)	1.42 ± 0.1 ^a^	1.3 ± 0.09 ^a^	1.4 ± 0.23 ^a^	2.3 ± 0.2 ^b^	1.55 ± 0.3 ^a^	2.5 ± 0.1 ^b^
Pentadecanoic (C16:1)	0.89 ± 0.0 ^a^	0.74 ± 0.09 ^a^	0.69 ± 0.03 ^a^	0.9 ± 0.14 ^b^	0.44 ± 0.1 ^a^	0.43 ± 0.06 ^a^
Pentadecanoic (C16:3)	1.0 ± 0.5 ^a^	0.59 ± 0.2 ^a^	0.9 ± 0.4 ^a^	0.52 ± 0.2 ^a^	0.73 ± 0.3 ^a^	0.6 ± 0.2 ^a^
Octadecanoic (C18:1)	3.73 ± 0. ^a^	3.96 ± 0.8 ^a^	2.97 ± 0.4 ^a^	4.45 ± 0.6 ^b^	3.46 ± 0.6 ^a^	4.48 ± 0.7 ^b^
Octadecanoic (C18:2)	21.5 ± 0.9 ^a^	24.2 ± 1.4 ^a^	25.1 ± 1.9 ^a^	22.5 ± 0.8 ^a^	24.2 ± 1 ^a^	20.8 ± 0.5 ^a^
Heptadecanoic (C18:3)	12.1 ± 3. ^a^	25.64 ± 2.3 ^b^	23.3 ± 3.3 ^a^	29.68 ± 6 ^b^	25.81 ± 4.1 ^a^	26.52 ± 0.4 ^a^
Heptadecanoic (C18:4)	1.0 ± 0.06 ^a^	1.27 ± 0.14 ^a^	1.04 ± 0.3 ^a^	1.2 ± 0.08 ^a^	1.13 ± 0.2 ^a^	1.16 ± 0.09 ^a^
Tetracosanoic (C20:3)	0.11 ± 0.01 ^a^	0.26 ± 0.0 ^b^	0.22 ± 0.07 ^a^	0.23 ± 0.04 ^a^	0.2 ± 0.06 ^a^	0.19 ± 0.04 ^a^

**Table 5 plants-10-02745-t005:** Antioxidant activity of control and endophytic bacterial-treated *Chenopodium ambrosoides, Chenopodium ficifolium,* and *Chenopodium botrys* sprouts. Data are represented by the means of at least 3 replicates ± standard deviations. Different small letter superscripts (a, b, and c) within a row indicate significant differences between control and endophytic bacterial samples.

	*C. ambrosoides*		*C. ficifolium*		*C. botrys*			
	Control	Endophyte	Control	Endophyte	Control	Endophyte	Extraction Solvent *	Cipro-Floxacin
Antioxidant Activities								
FRAP (mg gallic acid/gFW)	16.2 ± 1.2 ^a^	37.5 ± 1 ^c^	18.3 ± 0.6 ^b^	33.06 ± 3 ^c^	11.5 ± 0.1 ^a^	21.4 ± 6 ^c^	0.5 ± 0	-
ABTS (µmol trolox/gFW)	2.0 ± 0.1 ^a^	1.9 ± 0.2 ^b^	1.37 ± 0.1 ^a^	1.9 ± 0.01 ^b^	0.93 ± 0.1 ^a^	2.2 ± 0.3 ^b^	0.7 ± 0	-
DPPH%	18.8 ± 0.7 ^a^	13.7 ± 1 ^a^	12.2 ± 1 ^a^	16.37 ± 3 ^b^	12.7 ± 1 ^a^	17.06 ± 3 ^b^	0.1 ± 0	-

* Ex. Solvent is the extraction solvent (80% of ethanol).

**Table 6 plants-10-02745-t006:** Antibacterial activity of control and endophytic bacterial-treated *Chenopodium ambrosoides, Chenopodium ficifolium,* and *Chenopodium botrys* sprouts. Data are represented by the means of at least 3 replicates ± standard deviations. Different small letter superscripts (a, b) within a row indicate significant differences between control and endophytic bacterial samples.

	*C. ambrosoides*		*C. ficifolium*		*C. botrys*			
	Control	Endo	Control	Endo	Control	Endo	Ex. Solvent	Cipro-Floxacin
Antibacterial Activities (Zone Inhibition, mm)								
*S. saprophyticus* ATCC 19701	14.3 ± 0.4 ^a^	25.5 ± 1 ^b^	16.5 ± 0.3 ^a^	20 ± 2.3 ^a^	19 ± 0.1 ^a^	22.5 ± 0.6 ^b^	1.5 ± 0.1	33.4 ± 2.3
*S. epidermidis* ATCC 12228	10.9 ± 0.3 ^a^	22.0 ± 1 ^b^	14.5 ± 0.9 ^a^	17.6 ± 2 ^ab^	16. ± 0.3 ^b^	19.92 ± 1 ^ab^	0.9 ± 0	23.6 ± 5
*E. faecalis* ATCC 10541	15.0 ± 0.3 ^a^	20 ± 0.9 ^b^	17.8 ± 0.4 ^a^	15 ± 1.7 ^ab^	16 ± 0.1 ^a^	19.1 ± 1 ^ab^	1.1 ± 0	21.5 ± 4.0
*S. salivarius* ATCC25975	13.2 ± 0.8 ^a^	17 ± 0.4 ^ab^	16.2 ± 0.3 ^a^	20 ± 0.4 ^a^	13 ± 0.4 ^a^	17.09 ± 1 ^ab^	0.89 ± 0	30.6 ± 2.4
*E. coli* ATCC 29998	14.52 ± 1 ^a^	18.3 ± 1 ^ab^	19.5 ± 0.6 ^a^	21 ± 0.7 ^a^	16 ± 1.0 ^a^	19.29 ± 1 ^ab^	1.2 ± 0	25.9 ± 0.7
*S.typhimurium* ATCC14028	16.2 ± 0.3 ^a^	20.6 ± 1 ^ab^	19.1 ± 0 ^a^	23 ± 0.6 ^b^	18 ± 0.2 ^a^	20.5 ± 1 ^ab^	0.5 ± 0	22.1 ± 0.6
*P. aeruginosa* ATCC10145	16.9 ± 0.5 ^a^	33 ± 1.6 ^b^	20.2 ± 0.7 ^a^	22 ± 1 ^a^	24 ± 0.2 ^a^	29 ± 1 ^b^	1 ± 0	22.8 ± 1 ^a^
*P. vulgaris* ATCC8427	16. ± 0.4 ^a^	26.3 ± 1 ^b^	18.7 ± 0.4 ^a^	17 ± 0.5 ^a^	20 ± 0.1 ^a^	23.9 ± 0.8 ^a^	1.9 ± 0.1	27.1 ± 1.5
*E.r aerogenes* ATCC 13048	15.4 ± 0.3 ^a^	28.6 ± 1 ^b^	17.8 ± 0.4 ^a^	22 ± 0.4 ^b^	21 ± 0.2 ^a^	25 ± 0.7 ^a^	0.7 ± 0.0	24 ± 0.7
*S. marcescens* ATCC99006	1.9 ± 0.06 ^a^	7.3 ± 0.6 ^b^	2.2 ± 0.05 ^a^	5.9 ± 0.5 ^b^	4 ± 0.05 ^a^	5.5 ± 0.3 ^a^	0.9 ± 0.1	8.1 ± 0.5
*S. typhimurium* ATCC14028	16 ± 0.5 ^a^	10 ± 0.5 ^ab^	19.2 ± 0.4 ^a^	17 ± 0.4 ^a^	12 ± 0.1 ^a^	13.89 ± 1 ^a^	1.1 ± 0	22 ± 0.2
*C. albicans* ATCC90028	6.9 ± 0.2 ^a^	9.8 ± 0.5 ^b^	8.0 ± 0.19 ^a^	7.9 ± 0.9 ^a^	8.1 ± 0.1 ^a^	9.3 ± 0.4 ^a^	0.07 ± 0	8.9 ± 0.4
*C. glabrata* ATCC90030	2.3 ± 0.07 ^a^	4.8 ± 0.2 ^b^	2.9 ± 0.14 ^a^	3.1 ± 0.1 ^a^	3 ± 0.03 ^a^	4.18 ± 0.1 ^a^	0.2 ± 0	4.0 ± 0.01
*A. flavus* ATCC9170	22 ± 0.5 ^a^	43 ± 1.1 ^b^	25.7 ± 0.6 ^a^	33.2 ± 5 ^b^	29 ± 0.3 ^a^	36 ± 1 ^b^	20.5 ± 1	33.29 ± 5 ^b^

**Table 7 plants-10-02745-t007:** Anti-inflammatory activity of control and endophytic bacterial-treated *Chenopodium ambrosoides, Chenopodium ficifolium,* and *Chenopodium botrys* sprouts. Data are represented by the means of at least 3 replicates ± standard deviations. Different small letter superscripts (a, b, and c) within a row indicate significant differences between control and endophytic bacterial samples.

	*C. ambrosoides*		*C. ficifolium*		*C. botrys*	
Anti-Inflammatory	Control	Endo	Control	Endo	Control	Endo
Cyclooxygenase-2 (µg/mL)	1.1 ± 0.2 ^a^	0.73 ± 0.0 ^ab^	1.2 ± 0.1 ^a^	0.6 ± 0.04 ^b^	1.3 ± 1 ^a^	0.45 ± 0 ^c^
Lipoxygenase (µg/mL)	7.2 ± 0.1 ^a^	5.7 ± 0.5 ^ab^	6.7 ± 4 ^a^	4.6 ± 0.6 ^b^	6.8 ± 0.7 ^a^	3.1 ± 0.6 ^c^

## Data Availability

Data presented in this study are available on reasonable request.

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
