# Peer review of "Bacterial Endophytes as a Promising Approach to Enhance the Growth and Accumulation of Bioactive Metabolites of Three Species of *Chenopodium* Sprouts"

_plants, 2021, doi:10.3390/plants10122745_

Round 1

Reviewer 1 Report

The presented paper concerns very interesting issue of endophytic bacteria's influence on the quality of Chenopodium sprouts. The Authors put much effort and work to prepare the presented experiments and I am very impressed by their idea and results:-) Howver, I have some small suggestions and questions to be improved and explained:

  • Introduction, lines 76-88: as the experiments were performed on Chenopodium sprouts, not the plant itself, I would recommend to rephrase the paragraph to present what is known about the sprouts - their phytochemical composition, nutrotional values etc., and also are Chenopodium sprouts popular for eating like fro example broccoli sprouts? Moreover, I would like to know if the Authors grown the sprouts from the three Chenopodium species for the very first time in the world or heve the sprouts from these species been grown earlier by other authors? If so, the obtained results should be referred ito those obtained by other authors in the discussion part
  • paragraph 2.1.1: what standards were used for HPLC analysis of phenolic compounds? Where were they bought? Moroever, in the results and discussion part the Authors should explain why they used these particular standards - was the presence of the compounds previously described in the tested Chnopodium species?
  • paragraph 2.1.2: where did the seeds come from? Were they bought? In what company? In line 169: the temperature of liquid nitrogen is not -80°C, please correct 
  • paragraph 3.7.2: in the title of the paragraph the word "sprouts" is missing

Author Response

Reviewer 1

The presented paper concerns very interesting issue of endophytic bacteria's influence on the quality of Chenopodium sprouts. The Authors put much effort and work to prepare the presented experiments and I am very impressed by their idea and results:-) Howver, I have some small suggestions and questions to be improved and explained:

  • Introduction, lines 76-88: as the experiments were performed on Chenopodium sprouts, not the plant itself, I would recommend to rephrase the paragraph to present what is known about the sprouts - their phytochemical composition, nutrotional values etc., and also are Chenopodium sprouts popular for eating like fro example broccoli sprouts?

Response: Thanks, more details are added

  • Moreover, I would like to know if the Authors grown the sprouts from the three Chenopodium species for the very first time in the world or heve the sprouts from these species been grown earlier by other authors? If so, the obtained results should be referred ito those obtained by other authors in the discussion part

Response: Chenopodium sprouts, especially Chenopodium quinoa, are known for their high nutritive value, being a source for proteins and carbohydrates, thus they could be eaten like other sprouts. Actually, we have grown the sprouts from the three Chenopodium species. We are not the first to apply sprouting, but this has been done on several plants. However, we are the first to use endophytic bacteria to improve the nutritional quality of sprouts, and this would be the novelty.   

  • paragraph 2.1.1: what standards were used for HPLC analysis of phenolic compounds? Where were they bought? Moroever, in the results and discussion part the Authors should explain why they used these particular standards - was the presence of the compounds previously described in the tested Chnopodium species?

Response: We used about 25 standards and bought them from Sigma Aldrich. The standards were chosen based on their availability. Also according to our survey, we have found that Chenopodium is rich in such phenolic compounds.

  • paragraph 2.1.2: where did the seeds come from? Were they bought? In what company? In line 169: the temperature of liquid nitrogen is not -80°C, please correct 

Response: we brought the seeds from Agricultural Research Centre, Giza, Egypt, and we added this information to the method section. The temperature of liquid nitrogen is – 196, and we corrected it.

  • paragraph 3.7.2: in the title of the paragraph the word "sprouts" is missing

Response: Thanks, done

Reviewer 2 Report

Dear Authors,

Why PAR of 400 was chosen during vegetation experiments? (Line 166)

Is it true that nitric acid was used as a standard in elemental analysis using ICP-MS?

Author Response

Reviewer 2

Dear Authors,

Why PAR of 400 was chosen during vegetation experiments? (Line 166)

Response: Thanks, we only apply low PAR (200)

Is it true that nitric acid was used as a standard in elemental analysis using ICP-MS?

Response: Thanks, we corrected it, it was used for extraction.

Reviewer 3 Report

Comments to authors

Major comments

1) You have the wrong template for the manuscript. It is for publication in Metabolites not Plants journal. Please use right template. In addition, you have to separate Results and Discussion sections according to the instructions for authors.

2) You have to rewrite and rephrase most of the M&M subsections because you have to many similarities with citing manuscripts (total of 39%). Especially subsections DNA extraction,  2.2.1, 2.2, 2.3, 2.4.3, 2.4.4., 2.4.5., whole 2.5 subsection and 2.6

3) You Figures have poor resolution. Please correct them according to the manuscript template.

4) There are inconsistencies related to the conducted analyses on bacterial strains and sprouts.

5) English grammar and spelling editing is required.

Introduction

Lines 78-88 The sentences are too long and repetitive. Split sentences and rephrase.

Lines 89-93 Split into two sentences.

Lines 93-106 You should rephrase and reorder sentences in this whole paragraph and make it clearer. Define aims and hypothesis of your work.

M&M section

Lines 110-111 remove as being fertile…

Did you isolate bacteria from all 3 Chenopodium species?

Lines 111-112 Rephrase sentence, check grammar

Use L instead l for units, g for gram

put space between number and time (30 s not 30s). Correct this in whole manuscript text.

Line 118 use mix well instead shacked

DNA extraction should be in a separate subsection.

Write down the size of the trays for plant growth. Were the all 3 Chenopodium species grown?

Write down how much fresh sprouts tissue did you use for each analysis.

LOX and COX  Please describe how did you obtain plant extract?

Results

Table 1. This is very confusing because the title of the table is related to the morphological and biochemical characterizations of bacterial isolates, yet you put also data of Enzyme and biological activities etc. These analyses were not described in 2.1.1 subsections for bacterial characterization but only for sprouts. Please explain. Also, units for all measured parameters are missing.  

Figure 2 Biomass g of what? Write Endo strain

Figure 3 write Endo strain

Author Response

Reviewer 3

Major comments

1) You have the wrong template for the manuscript. It is for publication in Metabolites not Plants journal. Please use right template. In addition, you have to separate Results and Discussion sections according to the instructions for authors.

Response: Thanks, we replace it with the right one. According to the journal instructions, the Results and Discussion sections can be also combined, being more compacted, and thus the results would be better linked to the discussion.    

2) You have to rewrite and rephrase most of the M&M subsections because you have to many similarities with citing manuscripts (total of 39%). Especially subsections DNA extraction,  2.2.1, 2.2, 2.3, 2.4.3, 2.4.4., 2.4.5., whole 2.5 subsection and 2.6

Response: Thanks, we carefully rephrased all methods

3) You Figures have poor resolution. Please correct them according to the manuscript template.

Response: Thanks, we improved the presentation of the figures

4) There are inconsistencies related to the conducted analyses on bacterial strains and sprouts.

Response: More details on the methods applied for bacterial analyses were added

5) English grammar and spelling editing is required.

Response: Thanks, we have critically checked the English language.  

Introduction

Lines 78-88 The sentences are too long and repetitive. Split sentences and rephrase.

Response: Thanks, done

Lines 89-93 Split into two sentences.

Response: Thanks, done

Lines 93-106 You should rephrase and reorder sentences in this whole paragraph and make it clearer. Define aims and hypothesis of your work.

Response: Thanks, done

M&M section

Lines 110-111 remove as being fertile…

Response: Thanks, done

Did you isolate bacteria from all 3 Chenopodium species?

Lines 111-112 Rephrase sentence, check grammar

Response: Thanks, done

Use L instead l for units, g for gram

Response: Thanks, done

put space between number and time (30 s not 30s). Correct this in whole manuscript text.

Response: Thanks, done

Line 118 use mix well instead shacked

Response: Thanks, done

DNA extraction should be in a separate subsection.

Response: Thanks, done

Write down the size of the trays for plant growth. Were the all 3 Chenopodium species grown?

Response: the size of each tray is (8*12*4), where the sprouts of each treatment were grown in different trays

Write down how much fresh sprouts tissue did you use for each analysis.

Response: about 200 mg fresh sprouts were taken

LOX and COX  Please describe how did you obtain plant extract?

Results

Table 1. This is very confusing because the title of the table is related to the morphological and biochemical characterizations of bacterial isolates, yet you put also data of Enzyme and biological activities etc. These analyses were not described in 2.1.1 subsections for bacterial characterization but only for sprouts. Please explain. Also, units for all measured parameters are missing.

Response: more details were added to this section.

Figure 2 Biomass g of what? Write Endo strain

Response: Biomass of whole sprouts

Figure 3 write Endo strain

Response: Corrected

Round 2

Reviewer 3 Report

Comments to Authors revision 2

I would appreciate if this time authors would carefully read all my suggestions and answer all my questions.

Lines 75-80 Check grammar, rephrase this paragraph.

Line 80-82 This sentence makes no sense, it is missing something

Line 176 Please add after conducted… with aim to….

Line 81-82 Please rephrase to:  We hypothesize that the use of bio-enhancer, such as endophytic bacteria, ….

Table 1 Add measurement units for each of the parameters in the table

Line 319 This part is repeating, rephrase

Line 333 Molecular characterization  should be in a separate subsection

Line 351 Total chlorophyll (Chl a+b)  has not changed significantly after treatment with Endo, please rephrase or specify

Table 2 Change in Chl Ab to Chl a+b or total chlorophyll

Figure 1 in axis title  g/plant g of what? % of what How do you explain no statistical difference in % of Moisture? Please add this in discussion.

Lines 401-419 This whole paragraph should be under the paragraph describing  photosynthesis and before biomass, it makes more sense. In this paragraph Line 406 you write that there is significant enhancement for chl ab (you mean total chlorophyll?) but his is not true, please rewrite

Line 409 use another word instead harmony

Line 427-428 Write down that % of increase in K content

Line 438 There is no increase of Zn content in your study

Lines 442-445 Is that the case in your study? Please relate your results to this citation.

Table 3  I suggest that your presentation of the results in this section follow data in the Table and manuscript text. So, either you reorder the presentation of the results in the text or in the Table 3.

Lines 452-456 Not all vitamins and not in all species!!! For example, riboflavin  and thiamin are only increased in C.  botrys. This is not true for Vit C!!!!  form your results Vit C  increased in all species compared to control. Please correct.

Line 457-458 From your results C. ficifolium has more Vit E in comparison to all other species. So, your claim is not true.

Line 461-462 This is not true.  From results in Table 3 Ch. ambrosoides also has increased total proteins compared to control not as much as other two species but it is statistically significant. Please rephrase this statement.

Line 463 Not true Fat content is increased in C. ficifolium  and C. botrys. Please correct.

Line 463-464 Why the moisture content is mentioned here? This should be related to results of biomass or? This has nothing to do with this results in table 3.

Lines 505-507  I don’t understand the point of this sentence?

Table 4 correct units for Amino acid biosynthesis enzymes

Lines 578-580 Check results in Table 4 and in the text. Oxalic acid is also elevated in C. ficifolium and botrys, while succinic acid only in C. botrys. Mallic acid is very similar in C. ambrosoides in control and treatment. Check!

Lines 613-618 Add sentence about measured Phenylalanine values in 3 Ch. Species.

Table 5 Check letters for FRAP assay (b and c ? I C. ficifolium. What does Ex. solvent mean? Units? Describe in table footnotes. ABTS was increased in C. ambrosoides according to the statistic in the Table ! Check your results in the Table!

Antibacterial activities should be in separate Table (Table 6) and linked to 2.7.1. subsection it will be easier for readers to follow. Units for Ciprofloxacin are  missing in the table. I don’t understand the letter b in last column next to A. flavus (values for Ciprofloxacin).

Lines 673-674 Your result also showed that C. ambrosoides showed antibacterial effect against E. faecalis point that out. Do not just list other peoples work. Compare your results to others.

Table 6 Please add measurement units for enzymes. Be consistent either use Endo or Endophyte in Tables and Figures

Lines 681-682 “remarkable reductions” for all sprouts? C. ambrosoides values for control and treatment are similar. Accordingly, check your claims in text related to this species.

M&M

3.1.1 Isolation of endophytes. Again, where they isolated from all 3 species. Define!

What about Moisture measurements? Describe? Define! Units!

Sprout fresh weight was expressed as Biomass later in  figure 2? Indicate that in this section!

Line 744 reference Williams et al 1983. I don’t see this reference in the list I want to check how did you measured antioxidative activities in endophytes. Or am I missing something? Did you measure this values in the total isolates or how? This is not clear to me. Last time you ignored this question. Please explain.

Line 757 About 25 standards? Standards of what!? What does it mean that they were chosen based on their availability?

Line 748-750 Remove this sentence: Also, according…

Line 751 spelling change spout to sprout

3.1.3. Specify and name all 3 species of Chenopodium that you used for research in this section.

Line 784 frozen not freez, and stored for further ….

Line 793 Change to: Supernatant was filtered and analyzed…

3.4.1. You missing references for lipid content and crude fiber analyses. Saline solution? How did you calculate lipid content.? The last sentence is missing something. It does not make sense. This part is very poorly written. Please rephrase it.

3.4.6. What about Vit C and E determination?

3.5.3 LOC and COX you ignored my question last time. Again, how did you obtain extracts for these analyses?

Conclusion should be more specific in regard to that specific strain that showed best results. Please include that. How about economic impact of using this bacterial strain for improving nutritional values of the Chenopodium sprouts, are these procedures low or high cost? Is it economically feasible?

Author Response

Dear Reviewer,

Thanks for the critical review, indeed incorporating your valuable comments improved our manuscript quality.  We carefully incorporated your comments and checked whole manuscript for improving the manuscript at presentation and English level.

Round 3

Reviewer 3 Report

Comments to Authors

Table 1 Please define units for Bioactive compounds production. Describe in table footnote what  + and – stands for. E.g. + present, - not present

Line 221 use in text chl a+b

Regarding the K content (lines 356-357) You did not understand what I meant. Since you wrote in the previous version that “The highest percentage  was recorded for K in both control and inoculated plants.” I wanted that you write exact value of that percentage. Also, check grammar.

Table 4 correct unit for Amino acid biosynthesis enzymes  (nmol/mg protein.min)

Figure 3 Correct the name of enzyme PAL in graph

Correct the text related to PAL activity  in Chenopodium species. Write down that the substrate is phenylalanine (line 527). It is not true that only significant increment was observed only for C. ficifolium, on a contrary only in this species PAL is not significantly increased compared to control!!!!

Table 5 check spelling in table foot note related to defining ex. Solvent

M&M section

Subsection 3.1.1

1) Add in this part of the text that the isolates were done only from C. ambrosoides

2) Quote: “Total antioxidant activity namely, ferric reducing antioxidant power (FRAP), 2-diphenyl-1-picrylhydrazyl (DPPH), and 2,2′-azino-bis(3-ethylbenzothiazoline-6- sulfonic acid) (ABTS) was done by grinding 30 mg freeze-dried bacterial cells in liquid nitrogen and extracted in 2 mL of ice-cold 80% ethanol. “

I really don’t understand how did you weighted 30 mg of bacterial cells? Or is this really beyond my comprehension! This is the most problematic part in the manuscript. Please, once again, describe in detail this part of experiment and add appropriate references. I took the time and effort to find protocol such as yours but, unfortunately, I wasn’t able to find it. I only found protocol for measurement of DDPH activity in bacterial cell suspension (Lin, Xiangna, et al. "Lactic acid bacteria with antioxidant activities alleviating oxidized oil induced hepatic injury in mice." Frontiers in microbiology 9 (2018): 2684.) Therefore, once more describe in detail how did you measure FRAP, DPPH and ABTS. This is important for experiment reproducibility and for readers to understand.

Subsection 3.5.3 Write which ethanol was used for extraction of spots (70%, 80% ???).

Write mL instead ml.

COX assay- write the name of manufacturer of the assay kit.

Author Response

Table 1 Please define units for Bioactive compounds production. Describe in table footnote what  + and – stands for. E.g. + present, - not present

Response: Thanks, we added units for Bioactive compounds, and also we indicated signs + and – for presence or absence

Line 221 use in text chl a+b

Response: Thanks, done

Regarding the K content (lines 356-357) You did not understand what I meant. Since you wrote in the previous version that “The highest percentage  was recorded for K in both control and inoculated plants.” I wanted that you write exact value of that percentage. Also, check grammar.

Response: Thanks, we added the percentages of increases in K content in comparison to control, and also we carefully checked grammar

Table 4 correct unit for Amino acid biosynthesis enzymes  (nmol/mg protein.min)

Response: Thanks, detailed unites were added to the table

Figure 3 Correct the name of enzyme PAL in graph

Response: Thanks, corrected

Correct the text related to PAL activity  in Chenopodium species. Write down that the substrate is phenylalanine (line 527). It is not true that only significant increment was observed only for C. ficifolium, on a contrary only in this species PAL is not significantly increased compared to control!!!!

Response: Thanks, we checked and corrected it

Table 5 check spelling in table foot note related to defining ex. Solvent

Response: Thanks, done

M&M section

Subsection 3.1.1

1) Add in this part of the text that the isolates were done only from C. ambrosoides

Response: Thanks, we added this

2) Quote: “Total antioxidant activity namely, ferric reducing antioxidant power (FRAP), 2-diphenyl-1-picrylhydrazyl (DPPH), and 2,2′-azino-bis(3-ethylbenzothiazoline-6- sulfonic acid) (ABTS) was done by grinding 30 mg freeze-dried bacterial cells in liquid nitrogen and extracted in 2 mL of ice-cold 80% ethanol. “

I really don’t understand how did you weighted 30 mg of bacterial cells? Or is this really beyond my comprehension! This is the most problematic part in the manuscript. Please, once again, describe in detail this part of experiment and add appropriate references. I took the time and effort to find protocol such as yours but, unfortunately, I wasn’t able to find it. I only found protocol for measurement of DDPH activity in bacterial cell suspension (Lin, Xiangna, et al. "Lactic acid bacteria with antioxidant activities alleviating oxidized oil induced hepatic injury in mice." Frontiers in microbiology 9 (2018): 2684.) Therefore, once more describe in detail how did you measure FRAP, DPPH and ABTS. This is important for experiment reproducibility and for readers to understand.

Response: Thanks for valuable comments, we added more detailed on bacterial cell peroration and extraction for antioxidant activity. For antioxidant activities measurement, the purified endophyte strain was cultured on Glycerol-Yeast Extract Agar (Glycerol 5 mL, Yeast extract 2 g, K2HPO4 1 g, Agar 15 g, Distilled water 1000 mL) plates at 28 °C for 8 days. Streptomyces strain was transferred to glycerol yeast-extract broth in Erlenmeyer flasks for propagation. The strain culture was incubated on a rotary shaker at 180 rpm, 28°C, for 8 days. Then, Streptomyces cells were collected by centrifugation

for 10 min at 8000g at 4°C for cell precipitation. After washing the cells with sterile normal saline solution, the suspension was freeze-dried and repeatedly extracted with ethanol. After removing the ethanol solvent using rotary vacuum evaporator at 37 °C, the final extract (30mg) was obtained and suspended in 10mL of 80% ethanol (3mg/mL). Total antioxidant activity namely, ferric reducing antioxidant power (FRAP), 2-diphenyl-1-picrylhydrazyl (DPPH), and 2,2′-azino-bis (3-ethylbenzothiazoline-6- sulfonic acid) (ABTS) were measured by using 80% ethanol extract (2mg/mL).

Moreover, the methods in details of antioxidant measurements are mentioned in detailed in section 3.5.1.

Subsection 3.5.3 Write which ethanol was used for extraction of spots (70%, 80% ???).

Response: We added that it was 80% ethanol

Write mL instead ml.

Response: Thanks, done

COX assay- write the name of manufacturer of the assay kit.

Response: Thanks, added

Round 4

Reviewer 3 Report

Comments to Authors

In this version author significantly improved problematic M&M sections.

There are still some minor corrections that need to be addressed:

1) In Title of the manuscript you put comma after metabolites. Delete it.

2) Table 4 you wrote units for GS, DHDPS an CGS as /mg protein. min. Is this correct? Or it should be /mg protein min-1?

3) Table 5 check spelling and grammar in table footnote related to *Extraction solvent ….

4) Lines 548 correct H2S formulae

5) Lines 550-551 correct formulae of chemical compounds and ml to mL. Check in all manuscript text

6) Line 557 put Streptomyces in italics

7) all references in text should be represented by number (see lines 545, 664) and add them in Reference list as well. Check in all manuscript text.

8) Check grammar and spelling. Check spaces between words, check font and paragraph space in text.

Author Response

Comments to Authors

In this version author significantly improved problematic M&M sections.

There are still some minor corrections that need to be addressed:

1) In Title of the manuscript you put comma after metabolites. Delete it.

Response: Thanks, done

2) Table 4 you wrote units for GS, DHDPS an CGS as /mg protein. min. Is this correct? Or it should be /mg protein min-1?

 Response: Thanks, we corrected it

3) Table 5 check spelling and grammar in table footnote related to *Extraction solvent ….

Response: Thanks, done

4) Lines 548 correct H2S formulae

Response: Thanks, done

5) Lines 550-551 correct formulae of chemical compounds and ml to mL. Check in all manuscript text

Response: Thanks, we corrected and checked formulae in all manuscript text

6) Line 557 put Streptomyces in italics

Response: Thanks, done

7) all references in text should be represented by number (see lines 545, 664) and add them in Reference list as well. Check in all manuscript text.

Response: we checked all references and put them by number

8) Check grammar and spelling. Check spaces between words, check font and paragraph space in text.

Response: we checked grammar and English language, and also spaces